# At-site and regional frequency analysis of extreme rainfall in Belgium based on radar estimates

Edouard Goudenhoofdt[1], Laurent Delobbe[1], and Patrick Willems[2]

[1]The Royal Meteorological Institute of Belgium, Brussels, Belgium
[2]Department of Civil Engineering - Hydraulics Division, University of Leuven, Leuven, Belgium

*Correspondence to:* Edouard Goudenhoofdt[edouard.goudenhoofdt@meteo.be]

**Abstract.** In Belgium, only rain gauge time-series have been used so far to study extreme rainfall at a given location. In this paper, the potential of a 12-year quantitative precipitation estimation (QPE) from a single weather radar is evaluated. For the period 2005-2016, 1 h and 24 h rainfall extremes from automatic rain gauges and collocated radar estimates are compared. The peak intensities are fitted to the exponential distribution using regression in QQ-plots with a threshold rank which minimises the mean squared error. A basic radar product used as reference exhibits unrealistic high extremes and is not suitable for extreme value analysis. For 24 h rainfall extremes, which occur partly in winter, the radar-based QPE needs a bias correction. A few missing events are caused by the wind drift associated with convective cells and strong radar signal attenuation. Differences between radar and gauge rainfall values are caused by spatial and temporal sampling, gauge underestimations and radar errors. Nonetheless the fit to the QPE data is within the confidence interval of the gauge fit, which remains large due to the short study period. A regional frequency analysis for 1 h duration is performed at the locations of 4 gauges with 1965-2008 records using the spatially independent QPE data in a circle of 20 km. The confidence interval of the radar fit, which is small due to the sample size, contains the gauge fit for the two closest stations from the radar. In Brussels, the radar extremes are significantly higher than the gauge rainfall extremes; but similar to these observed by an automatic gauge during the same period. The extreme statistics exhibit slight variations related to topography. The radar-based extreme value analysis can be extended to other durations.

# 1 Introduction

Localised rainfall extremes can have a strong impact on human activities especially in urban areas (Ootegem et al., 2016). For flood management applications (e.g. sewer system and dam design) it is needed to know the probability that rainfall exceeds a given amount. This probability is often expressed as the rainfall level which, on average, will be exceeded once over a given period of T years, which is defined as the return period. For infrastructure design application, one is interested in return periods from 50 to 100 years. Such long return periods often exceeds the available observation period and a model is needed.

Extreme values are often extracted from a time series using block maxima, typically over one year (AM) for meteorological data. The performance of the statistical modelling applied to AM data is limited by the number of years available. The peak-over-threshold (POT) method, where values exceeding a given threshold are kept, allows to increase the number of samples. The extreme value theory showed that for independent random variables, AM and POT series converge asymptotically to the 3-parameters distributions known as GEV and GPD, respectively.

Different fitting methods to the extreme value distributions have been developed in the literature. The maximum likelihood estimator (MLE) is the most widely used fitting method but for small samples it can lead to unrealistic parameter estimates. This problem is partially addressed with the generalised MLE proposed by Martins and Stedinger (2000) or the L-moments method (Overeem et al., 2009). The above methods do not focus on the tail of the distribution, which is the most relevant for risk analysis. For this goal, Willems et al. (2007) proposed a method based on regression in Q-Q plots.

To reduce the uncertainty associated with the limited number of data at a single site, regional frequency analysis (RFA) methods have been proposed (Svensson and Jones, 2010). The RFA is characterised by the selection of the regions and the parameter estimation approach applied to each region (Buishand, 1991). There are numerous studies of RFA for rainfall extremes based on rain gauge datasets. The index flood approach, which considers that only the location parameter varies in the region, is very popular (Gellens, 2000; Sveinsson et al., 2001; Rulfova et al., 2014). Uboldi et al. (2014) used a bootstrap technique to randomly select data from neighbouring locations with a probability depending on the distance and altitude difference with the target location. The combined use of POT and RFA methods is recommended by Roth et al. (2015).

One of the challenges in RFA is the intersite dependence (e.g., Hosking and Wallis, 1988). Even for 1 h duration, rainfall maxima exhibit spatial correlation (Vannitsem and Naveau, 2007). Using the sum of the length of all sites is common but causes underestimation of the extremes (e.g., Bardet et al., 2011). Several approaches have been proposed to deal with this problem (e.g., Castellarin, 2007; Weiss et al., 2014).

To obtain the rainfall statistics at any given point, spatial models have been developed using geographical and climatological covariates (e.g., Cooley et al., 2007). In Belgium, Van de Vyver (2012) derived a spatial GEV model depending linearly on the altitude. Rulfova et al. (2014) found for 6 h rainfall in the Czech Republic that the assumption of a linear model might be too restrictive, especially for convective precipitation.

The rain gauge network can capture rainfall extremes for widespread situations. However, they can only catch a small part of rainfall extremes caused by convective storms, which exhibit strong spatial variations over short distances. The use of high resolution gridded rainfall datasets to study rainfall extremes is still in its infancy. This can be explained by their unavailability,

their processing requirements and their limited quality. Precipitation estimations from satellite offer global and relatively long records suitable for extreme value analysis (Marra et al., 2017) but still suffer from large uncertainties (Sapiano and Arkin, 2009). The best potential is currently provided by radar-based quantitative precipitation estimation (QPE) products. It should be noted that the radar estimates represent the averaged precipitation over a given area (typically a square of 1 km). While this

area is much bigger than the gauge area, we will consider it as representative for small scale precipitation. It has been shown that the sub-pixel variability of rainfall extremes is significant, especially for short durations (Peleg et al., 2016). The relatively short record of radar datasets is an issue if the extreme statistics depend only on time (i.e. are completely dependent spatially). While this is a reasonable assumption for larger duration (e.g. 1 day), it is difficult to prove for short duration (e.g. 1 h). In case of significant climatic variations, a short record will be more representative of the extreme statistics.

In a pioneer work, Overeem et al. (2009) showed that a 11-year radar data set is suitable to derive depth-duration-frequency (DDF) curves for the Netherlands. But some differences with rain gauge results were found for short durations. Based on a unique 23-year radar data set in Israel, Marra and Morin (2015) found that the DDF curves were generally overestimated but 60 % of them lay within the raingauge DDF confidence intervals. In Ontario (Canada), Paixao et al. (2015) demonstrate the potential to integrate radar (Digital Precipitation Array product) to rain gauge analysis, especially to identify homogeneous

regions of extreme rainfall. Saito and Matsuyama (2015) used a 26-year radar-gauge dataset (without RFA) to study the spatial variation of hourly rainfall extremes in Japan. They found significant spatial patterns but also large uncertainties in the radar datasets. Different index flood approaches were tested by Eldardiry et al. (2015) in Louisiana, who defined a region as a square window of 44 km size. They found for Louisiana (USA) that the relatively short period (13 years) explains the high uncertainty of the analysis, that the index flood method is recommended and that a systematic underestimation is associated with the radar

products (its spatial resolution is $4 \times 4\,km$). Haberlandt and Berndt (2016) found that the operational DWD product is only suitable for studies on longer durations after bias correction. Using a 10-year high resolution radar rainfall dataset, Wright et al. (2014b) performed a regional frequency analysis using stochastic storm transposition. They found that the radar-based IDF estimates generally reproduce conventional gauge-based IDF estimates but overestimate these for longer return periods and shorter durations.

The potential of the radar data can be fully exploited by studying the extremes of the mean (or maximum) rainfall over areas. With the goal of deriving alert thresholds for 159 regions in Switserland, Panziera et al. (2016) studied the areal rainfall maxima (with sizes from the pixel to the region). Using RFA on squares, Overeem et al. (2010) derived areal rainfall depth-duration-frequency curves for the Netherlands. Wright et al. (2014b) applied a similar methodology but on different catchments in Louisiana.

In this study, we want to demonstrate the potential of high-resolution radar-based QPE to derive rainfall extreme statistics at a given location. To our knowledge none of the previous studies combine a high quality radar-based QPE with a high quality reference rain gauge measurements. At the Royal Meteorological Institute of Belgium (RMIB), a QPE has been derived from the reprocessing of raw volumetric radar measurements. This dataset has been used for various applications such as case studies and model verification. The methodology to derive this dataset has been verified for the period 2005-2014 against an

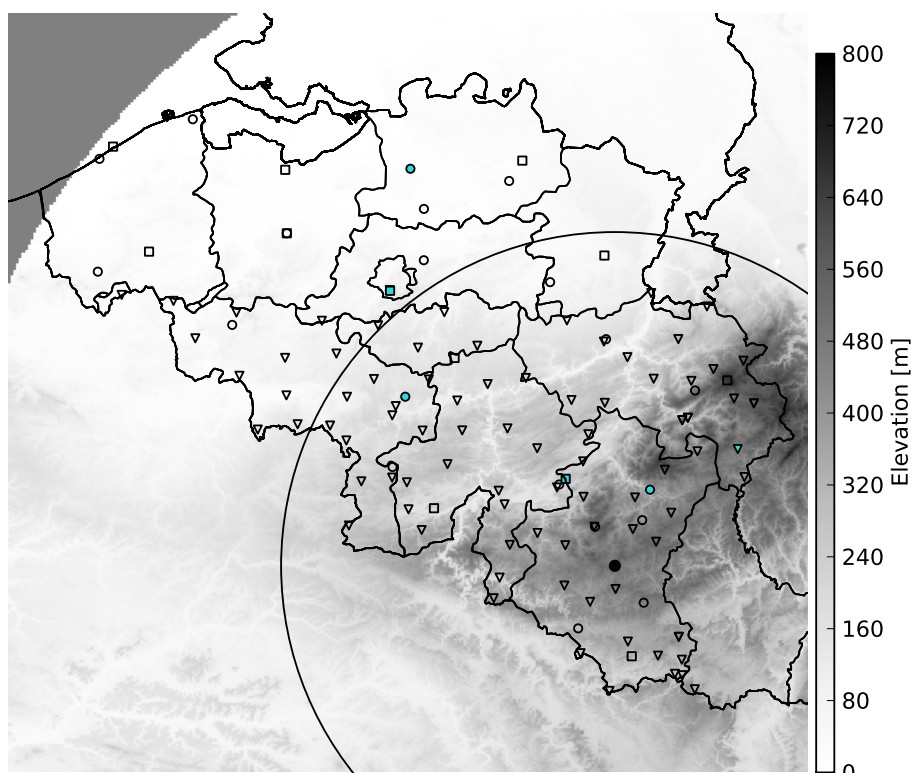

**Figure 1.** Elevation map centered on Belgium with the Wideumont radar (black dot) covering 240 km range (the black circle denotes the 120 km range) with AWS (square), SPW (triangle) and BUL (circle) rain gauge networks. The gauge locations selected in this paper are in cyan. Country borders with France, Luxembourg, Germany and the Netherlands are also displayed.

independent rain gauge network (Goudenhoofdt and Delobbe, 2016). RMIB also has a unique 40 year dataset of 10-min rain gauge measurements which has been used in extreme value studies (Vannitsem and Naveau, 2007; Van de Vyver, 2012).

Unlike existing radar studies, we select our data using the POT approach and use the QQR fitting method. Radar-based extreme statistics for 1 h and 24 h duration are compared with the ones derived from rain gauge data covering the same period. We propose a new regional frequency analysis which makes use of independent radar data in a predefined neighborhood. The results are compared with those obtained using the long-term rain gauge network. Finally, the regional approach is applied at each radar pixel on the whole of Belgium to study the spatial variations of the rainfall extremes.

## 2 Rainfall data

### 2.1 Raingauge measurements

Over the years, Belgium (Fig. 1) has been covered by several raingauge networks for different purposes.

Since the end of the 19th century, RMIB maintains a network (CLIM) of non-recording rain gauges from which rainfall measurements are taken at 8 am LT. The data are carefully controlled and used for climate applications (Journée et al., 2015).

A Hellmann-Fuess pluviograph has been in operation in Uccle (RMIB) from 1898 to 2008 and has enabled the compilation of a continuous time series of 10 min rainfall (Demarée, Gaston, 2003). The 10 min rainfall values had to be manually extracted from line graphs on papers. Starting from the fifties, additional rain gauges were installed to constitute a network (BUL) for hydrological research. Since the rain gauges underestimate the rainfall by 5-10% due to its mechanism, its records have been calibrated using a collocated gauge from the CLIM network.

For weather forecast purposes, the RMIB maintains a network of automatic weather stations (AWS) in Belgium. These stations provide rainfall measurements at 10 min temporal resolution. The tipping-bucket gauges are progressively replaced by weighted gauges (the first one was installed in Uccle on 10 February 2009). The data are available since 2002-2004 and have been quality controlled.

The hydrological service of the Walloon Region (SPW) maintains a dense network of hourly (every 5 min since 2012) rainfall measurements. The tipping bucket gauges are progressively replaced by weighting gauges since 2015. The data have been quality controlled by RMIB since April 2004.

It is important to know the limitations of the respective rain gauges in case of extreme rainfall. It is known (Nystuen, 1999; Duchon and Biddle, 2010) that tipping buckets underestimate high rainfall rates. The use of weighting gauges for extreme rainfall is discussed in Colli et al. (2012). Every 10 mm, the pluviograph has to be emptied which results in an underestimation in case of extreme rainfall. The calibration of the pluviograph is probably not sufficient for sub-daily extremes. Finally, the quality controls, albeit conscientious, can never be considered as perfect.

## 2.2 Radar estimation

The quantitative precipitation estimation (QPE) available on a 1 km grid every 5 min is made using an elaborated processing chain from the radar volumetric reflectivity measurements. The quality of the radar volume is controlled using several algorithms :

- a static clutter map : pixels with unrealistic high probability of rainfall are identified as clutter

- a beam blockage map : the percentage of the beam blocked by topography is computed using a simple propagation model

- a first clutter identification based on reflectivity differences between radar beam elevations

- a second clutter identification based on strong deviations of a pixel from its neighborhood and unrealistic lines

- a third clutter identification for radar echoes in cloud free areas determined by satellite observations

A maximum threshold for reflectivity is set to 55 dBZ to mitigate higher reflectivity values due to hail. The rainfall rate estimates are obtained using stratiform-convective classification, a 40 min averaged vertical profile of reflectivity (VPR), a bright band identification and a specific transformation to rain rates for the two precipitation regimes. The detailed procedure

is described in Goudenhoofdt and Delobbe (2016). As a reference for the QPE product, the CAP product is defined as the interpolation at 800 m above the radar level. It makes use of a standard Z R relationship, which comes from the hypothesis that the drop size distribution follows the distribution of Marshall-Palmer, as discussed in (Uijlenhoet and Pomeroy, 2001).

Consecutive rainrate estimates are integrated to obtain 10 min accumulations (5 min gaps are tolerated) to match the lowest resolution of the rain gauge data. Hourly accumulations are combined with the SPW gauges using a mean field bias correction. This method applied to the QPE product is referred to as the MFB product from now on. A more complex merging method (i.e. external drift Kriging) was tested but found to be unstable for some time moments.

It is important to mention the limitations of the radar products in case of extreme precipitation. The most important impact of the QPE processing on extreme values is the 55 dBZ reflectivity threshold used to mitigate hail. Using the convective Z R relationship, this corresponds to a maximum rainfall rate of 80 mm/hour. Higher values of about 100 mm/hour are possible when the standard Z R relationship is used for stratiform areas. This can only happen close to the radar where convective precipitation can not be identified. Slightly higher thresholds have been used by Overeem et al. (2009) (100 mm/hour) and (Wright et al., 2014b) (105 mm/hour). A higher threshold is used by Marra and Morin (2015) (150 mm/hour) but for a Mediterranean climate. Only half of the AWS gauges recorded up to 3 times more than 100 mm/hour in 10 minutes. Given the sub-pixel spatial variability, one can assume that this threshold will never be exceeded for the pixel average. This threshold can only partly correct for the overestimation due to hail. The second most important error is related to signal attenuation especially in case of well organised convective systems. This is why extremes might be underestimated the further the distance from the radar. In addition, the increasing radar sample volume will produce an underestimation of small scale extremes. The uncertainty in the Z-R relation is another important source of error.

## 2.3 Comparison framework

In this study, we will only consider validated rain gauge data. Given that the SPW network is used for merging, the radar dataset for 2005-2016 is used. To perform a direct comparison, the gauge data of AWS and SPW for the same period are used. For comparison against the reference BUL network, the gauge data for the period 1965-2010 are used. The timeseries of the BUL and CLIM networks have been tested for homogeneity by Van de Vyver (2012) and a selection of useful stations has been made. Gellens (2000) and Vannitsem and Naveau (2007) found that the vast majority of the CLIM and BUL time series are stationary for summer rainfall. However, the existence of a multi-decadal oscillation in rainfall extremes has been found in the Uccle time series (Ntegeka and Willems, 2008; Willems, 2013).

The 10 min rainfall accumulation from the gauge networks (AWS, BUL) and radar products (CAP, QPE) are summed to obtain sliding 1 h rainfall accumulations. Such duration is associated with convective storms, which can only be properly seen on radar images. The hourly bias obtained by the MFB method could be applied to the 10 min accumulations. However, it will not be used due to the possible risk of representativity errors related to convective storms and the small benefits expected.

The hourly rainfall from the SPW network and the radar products (CAP, QPE, MFB) are summed to obtain sliding 24 h rainfall accumulations. The SPW network is preferred to the AWS network because it is denser and more homogeneous. Such duration is mainly associated with widespread precipitation for which the benefit of merging methods is clear. The risk of

instability with MFB (e.g., in case of strong spatial variation of the bias) is tolerated given the significant expected benefit for widespread precipitation events.

It should be noted that using the lowest available duration for each network would result in an underestimation of the extremes due to the discrete time sampling (Marra and Morin, 2015). Additionally, random errors and time sampling difference can be compensated by performing the sum. For both the radar and the gauge, no missing data is tolerated in the sum to avoid underestimation. Furthermore, only timestamps with both radar and gauge data are kept.

Due to the amount of stations, it is not possible to analyse in details the results at each station. Therefore a few stations are picked at different distances from the radar (see Tab. 1 and Fig. 1). The Uccle station is chosen because it is included in the 3 networks, which makes inter comparison possible. The availability of the 1 h accumulation data is about 95 % for the radar products and close to 100 % for the AWS gauges. The radar availability of the 24h accumulation is lower than the 1 h accumulation because a significant part of the intervals without data are short. The availability of the SPW gauges is around 90 % but this is mainly due to the removal of snow events, when no extreme precipitation is expected. The availability of the BUL stations for the period 1965-2010 is highest at Uccle with 96.3 %, then about 86 % at Deurne and Gosselies. The station of Nadrin has only 60 % of availability (for the period 1965-2010) because it was started in 1978.

# 3 At-site frequency analysis

## 3.1 Methodology

It has been shown by Pickands III (1975) that the extreme values converge asymptotically to a generalized Pareto Distribution (GPD) :

$$F_{(\xi,\mu,\sigma)}(x) = \begin{cases} 1 - \left(1 + \frac{\xi(x-\mu)}{\sigma}\right)^{-1/\xi} & \text{for } \xi \neq 0, \\ 1 - \exp\left(-\frac{x-\mu}{\sigma}\right) & \text{for } \xi = 0. \end{cases} \tag{1}$$

with $\xi$, $\mu$ and $\sigma$ commonly defined as the shape, location and scale parameters. The special case when the shape parameter is equal to zero is defined as the Exponential distribution (EXP).

The choice of the threshold has an important impact on the estimation of the distribution parameters. When the number of selected values increases, the variance naturally decreases but the bias increases (due to the deviation from the theoretical distribution). It is more practical to use a threshold rank instead of a threshold value to control the sample size.

To apply the theory, the extreme values have to be independent but successive peaks within the same time window can be observed due to the nature of precipitation. For 1 h duration, two peaks are considered dependent if the time interval is less than 12 h as proposed by Ntegeka and Willems (2008). This choice is consistent with the characteristics of convective storms analysed in Goudenhoofdt and Delobbe (2013). Jakob et al. (2011) used a separation time of 24 h but found little sensitivity when taking lower or higher values. We also found that using 3 days hardly affects the selection of the 1 h extremes. For 24 h duration, we use a time interval of 3 days which is the typical scale of synoptic regimes. These choices are consistent with

Roth et al. (2014) who found empirically a temporal dependence of 1 day and 2 days for winter and summer precipitation, respectively. In practice, a peak is kept if it is the maximum compared to its dependent peaks (if any).

The type of the distribution can be derived by looking for the QQ-plot where the extremes behave in an asymptotic linear way. (Willems, 2000) found for the Uccle series that the tail of the distribution has an exponential behavior for all durations. In the gauge datasets used in this study, we also found a tendency for the EXP distribution. The EXP distribution is preferred for short period since estimating the shape parameter is very uncertain. Blanchet et al. (2015) found that GPD fails to robustly estimate the tail of the distribution because of lack of data and unrealistic return levels for very long return periods (when the shape parameter is positive). An additional argument for the EXP model is that it is less affected by observational errors, which plays an important role here.

In this study we use a fitting method based on regression in Q-Q plots (QQR) proposed by Willems et al. (2007). The Exponential Q–Q plot is the extremes $x$ versus minus the logarithmically transformed exceedance probability $1 - G(x)$. The EXP distribution appears as a line in this plot, with slope equal to the scale parameter $\sigma$:

$$x = x_t - \sigma ln(1 - G(x)) \tag{2}$$

where $x_t$ is the threshold level. The same properties hold for the plot of the return level $x_T$ against the return period $T$ when the latter is plotted on a logarithmic scale :

$$x_T = x_t + \sigma ln(T * M/n) \tag{3}$$

where $M$ is the number of extremes and $n$ the length of the timeseries.

The estimators for the slope are based on linear regression in the Q–Q plot above the specific threshold level $x_t$. Amongst the available estimators for $\sigma$ we used an unconstrained and unweighted linear regression.

The optimal threshold rank $t$ is found by minimization of the mean squared error (MSE) of the calibration. With our datasets, this rank is chosen between 18 and 30 considering the uncertainties and the relatively short period, respectively. Confidence intervals for the scale parameter are computed using a parametric bootstrap technique. The fitted distribution is used to generate 1000 extreme values series with a size corresponding to the optimal rank. The fitting procedure is applied to each of the 1000 series to obtain 1000 simulated scale parameters. The 10 and 90 percentiles of the simulated parameters are taken as the 10 % and 90 % confidence interval bounds for the true scale parameter.

### 3.2 Comparison of 1h extremes

The extreme events as seen by both the radar and the gauge are compared in table 2. Since the focus is on the tail of the distribution, only the 10 highest values from either the gauge or the radar data are selected. The events for which the probability of hail is high (i.e. when the threshold was applied) are highlighted. An event is considered as problematic if the corresponding radar or gauge extreme rank is below 30. For these events, the underlying precipitation patterns are analysed using the radar images. This comparison allows identifying the weaknesses of the gauge and radar datasets.

The maximum at Humain has been observed by both the radar and the gauge on 7 June 2016. This relatively high value can be due to randomness and the short period of records. But it is also possible that the other quantiles are underestimated

(the maximum was recorded by the new weighted gauge). There is generally a good match between the radar and the gauge quantiles except for the following events :

  – event 2 : the radar underestimates globally

  – event 7 : the gauge is located at the boundary of the convective cell

  – event 11 : the radar signal is strongly attenuated by a mesoscale convective system.

  – event 13 : there was probably snow in the gauge

  – event 14 : the gauge is located at the boundary of a convective cell.

The second highest quantile at Uccle has been observed by both the radar and the gauge on the 7th of October 2009. There is generally a good match between the two datasets. A few events are problematic :

  – event 1,4 : the gauge is at the boundary of a cell

  – event 9 : there is a stationary storm underestimated by the gauge

  – event 10 : the gauge is at the boundary of a cell and the radar is attenuated (same as event 2 in Humain)

  – event 11 : the radar signal is strongly attenuated (same as event 11 in Humain)

  – event 13 : the radar is attenuated

The problems with cell boundaries are easily explained : the radar estimation is taken at a given height above ground and the rain is subject to wind drift before reaching the ground. This effect increases with the distance to the radar. Due to its randomness, it should not affect the statistics. The other problematic events can be considered as missing data. Since the level of missingness is limited, the impact on the statistics is expected to be small.

Figure 2 shows the results of the extreme value analysis for 1 h rainfall accumulation. The return levels are obtained using formulas from Willems et al. (2007) which are based on the Weibull plotting position. Numerical values of the temporal independence, the optimal rank, the location parameter and the scale parameter can be found in table 3. The percentage of independent peaks (among peaks exceeding the threshold) is around 20 % for both the radar and the gauges at the two locations. This low value is mainly due to the fact that 5 consecutive values at 10 min resolution are correlated.

The empirical quantiles of the QPE product are systematically slightly lower than those for the AWS gauges. This may be expected as we compare point rainfall observations with rainfall averaged on a 1 km square. However, the underestimation of very high rainfall rate by tipping-bucket gauges can compensate for this effect. One also notes small groups of similar values for both the radar and the gauge, which are mainly associated with hail events. This can be explained by the effect of hail threshold and the rainfall rate limit, respectively. The extremes tend to be heavy tailed but this can be at least partially explained by the observation biases described above.

The fit of the EXP distribution is relatively good for the two locations with a relatively low MSE (not shown). The scale parameter tends to be higher for the gauge data than the radar data. In general, the uncertainty for the scale parameter remains high and this results in wide confidence intervals for higher return periods.

When using the CAP product, the higher quantiles are overestimated especially for Uccle. This can be mainly attributed to the effect of hail. This results in an overestimation of the scale parameter.

### 3.3 Comparison of 24h extremes

The comparison of the 10 highest extremes from either the radar (MFB) or the gauge (SPW) can be seen in table 4. For Uccle, most extreme values occurred during summer and are therefore associated with convective storms. There is a good match between the gauge and the radar except for a few events:

– event 8, 11 : the gauge is at the boundary of a convective cell

– event 13 : strong radar attenuation by a mesoscale convective system

– event 14 : snow episode probably underestimated by the radar

For Saint-Vith, the extreme values occurred either in summer or in winter with therefore a mix of convective and widespread precipitation episodes. The match is very good except for the following events :

– event 2 : at the boundary of a cell (probably with hail)

– event 3 : slight overestimation due to snow melting (QPE) ; overestimation due to non-uniform bias (MFB)

– event 13 : at the boundary of a cell

The problematic events not related to boundary effects can be considered as missing data. Since they are limited it is expected that they only slightly affect the statistics.

Figure 3 shows the results of the extreme value analysis for the 24 h rainfall accumulation. Numerical values can be found in table 5. The percentage of independent peaks (amongst peaks exceeding the threshold) is between 6 % and 9 % for the two locations and for all datasets. This is what we expect from 24 h accumulation available every hour.

For Uccle there are not many differences between QPE and MFB because most events are associated with convective storms. Compared to the gauge quantiles, the radar quantiles are lower below 1-year and higher between 1-year and 5-year return periods. This can be attributed mainly to hail overestimation by the radar and gauge losses. It results in a higher scale for the radar, which is close the upper bound of the gauge confidence interval.

For Saint-Vith, there is a clear effect of the bias correction (MFB) to remove the underestimation of the QPE product. As for Uccle, the radar quantiles are higher for return periods higher than 2 years but the effect is limited because less convective storms are involved. The final result is a good match of the two distributions for this station.

For the two stations, no significant instability in the MFB values have been found.

For Uccle, the CAP product overestimates the scale parameter and underestimates the location parameter due to hail and VPR errors, respectively. For Saint-Vith, the quantiles (not shown) are similar to QPE except for a very high unrealistic maximum.

## 4 Regional frequency analysis

### 4.1 Methodology

As in Overeem et al. (2009) and Wright et al. (2014b) we consider that the extreme statistics are the same within the region. The region should be sufficiently large to have a large sample size (many extremes) and small enough to neglect extreme statistics variability. No strong variability is expected in Belgium because it is a relatively flat country. Therefore we define our region as the radius of 20 km around the target location. A similar size has been used in other radar studies (e.g., Overeem et al., 2009; Wright et al., 2014b; Eldardiry et al., 2015).

We also consider that the extremes observed within the 20 km radius during a time window of 12 h are dependent. As in Wright et al. (2014b), we keep only the maximum amongst dependent values. We therefore implicitly assume that the regional maximum follows the same distribution as the local extremes. The possible benefit of taking one extreme value at random is an open question. It is important to remind that we are interested in the extreme statistics of any given pixel in the region. This is different from studying the extreme statistics of the maximum rainfall over the region as in Panziera et al. (2016). We also tested the hypothesis that 1 hour extremes are independent after a certain distance which is set to 10 km. This distance corresponds to the maximum expected size of a convective cell (Goudenhoofdt and Delobbe, 2013). If this is true it allows to reduce the uncertainty of the analysis. In the text, we will refer to these datasets by the names RFA and R10, respectively.

Due to the spatial dependence, the effective length $n_{eff}$ of the pooled time-series is smaller than the total length of the records. The total length is obtained by multiplying the number of years $n$ by the number of pixels $N$ :

$$n_{max} = n \times N. \tag{4}$$

In this study $n_{eff}$ is computed by multiplying $n_{max}$ by the fraction of spatially independent peaks, amongst peaks exceeding the threshold. The latter is obtained by dividing the number of independent peaks by the total number of peaks. It can be shown that this is the same as the method based on the averaged exceedence rate found in Wright et al. (2014b) and explained in details by Weiss et al. (2014). The large number of peaks available from the radar data allows us to choose a higher threshold rank. This increase in sample size leads to a more reliable extreme value analysis, which is the final goal of this radar-based RFA. Accordingly the QQR method is applied for threshold ranks between 30 and 100 and the optimal rank is found.

### 4.2 Comparison with rain gauges

Figure 4 and 5 shows the results of the regional frequency analysis for 1 h rainfall accumulation at the 4 locations selected from the BUL network. The results of the at-site frequency analysis for the gauge and collocated radar pixels are showed as reference. Numerical values can be found in table 6. The percentage of temporally independent extremes for the gauge is close to 30 % for Deurne and Uccle while it is slightly above 20 % for the two others stations. This suggests that there are

larger clusters which might be related to altitude. Above the threshold, the percentage of spatially independent extremes (RFA) ranges from 1.1 % (Uccle) to 2.6 % (Nadrin). The effective period length of the pooled dataset is then between 200 and 500 years. Using a decorrelation distance of 10 km results in twice more data, which is more than one expects from randomness. It suggests that convection can be organised at large spatial scales.

5     The radar images associated with each maximum of the radar-based RFA is analysed :

- Deurne and Uccle (28 July 2006) : several supercells on the whole of Belgium

- Gosselies (22 August 2011) : a squall line moving parallel to the flow

- Nadrin (26 July 2008) : a stationary convective cell

The highest extremes exhibit abrupt variations in the form of steps for both the gauge and radar. This could be explained by 10  the siphonage of the gauge and hail threshold, respectively. Since Nadrin is close to the radar, the standard Z-R relationship is used instead of the convective Z-R relationship. This permits higher rain rates (i.e. 100 mm/hour).

    The gauge extremes are relatively low at Deurne and Uccle compared to Nadrin and Gosselies. The radar extremes are lower at Deurne compared to the other stations. This can be at least partially attributed to the large sample volume at this range. The match between the gauge and the radar (RFA and R10) is good except at Uccle with much higher radar extremes. The 15  RFA exhibits higher extremes than R10 suggesting some dependence beyond 10 km. Indeed the results should be similar if the hypothesis of independence after 10 km was valid.

    This can be partially attributed to hail but the similar 4 highest extremes suggest a gauge limitation. It is also striking that half of the 20 highest gauge extremes occurred during the period 1999-2008 (not shown). This positive trend for Uccle is possibly related to the urban heat island effect (Hamdi and Van de Vyver, 2011). The uncertainty of the radar fit is low because of the 20  larger sample size, due to which a higher rank can be chosen. Furthermore, the fit is less impacted by the potentially large errors of the highest extremes. The location parameter (corresponding to the threshold) increases with the sample size of the products.

    Except for the Uccle station, the scale parameter is the lowest for the QPE dataset due to the bias as a result of the small sample size. The scale parameter of the pooled radar datasets is slightly higher at Deurne and significantly higher in Uccle. For 25  Gosselies and Nadrin, the R10 and BUL data have similar scales while it is slightly higher for the RFA data. The fit to the RFA and R10 data is within the uncertainty bound of the fit to the BUL data. For those two stations, the fit to the BUL data is even in the small uncertainty bound of the fit to the RFA data.

## 4.3   Spatial maps

We apply the regional frequency analysis described above for 1 hour duration to all pixel locations in Belgium with some 30  modifications. We use a smaller radius of 10 km to reduce the computation cost and consider that all pixels are spatially dependent. This smaller radius improves the resolution of the maps at the expense of a higher uncertainty. Several pixels in the radar dataset are affected by permanent non-meteorological echoes. They can be identified by an unrealistic high frequency of

extremes. In practice one looks at the distribution of the number of values exceeding 12 mm/hour. The pixels with more than 50 exceedances have been found as outliers and removed. To make the comparison easier, we choose a fixed threshold rank of 60. No larger ranks have been considered due to computational limitations.

Figure 6 shows the results of the regional frequency analysis applied to Belgium. The provinces of Belgium are also displayed to help comparison between the maps. No values are shown beyond the 180 km range because the quality of the radar QPE is significantly reduced. The return periods are computed using equation 3 and therefore depends on the scale parameter and the effective length. The higher the scale the higher the difference between the 10-year and 100-year return levels.

Some artifacts due to the radar and the regional approach can be seen on the maps. The effective length decreases significantly beyond 100 km meaning that the spatial dependence increases. This is due to the fact that the actual radar sample is larger than the 1 km pixel at those ranges. Circular patterns appear on the maps due to the influence of the pixels located at their centers. The high values are caused by pixels contaminated by non-meteorological echoes (e.g. at the German border) and hail. A stronger filter for non-meteorological echoes is not used because it could remove actual precipitation information. The circular effect might be reduced by using a larger radius or a higher threshold rank but this is computationally expensive. Areas with a 10-year return level exceeding 30 mm are mainly located beyond 100 km. This is probably due to an increased contamination by hail with the distance to the radar (and the height of the measurements). The small scale variability in the study area can be explained by uncertainties due to the sample size.

There is some correlation between the 10-year return level and the scale parameter. Therefore the spatial patterns between the two return periods are similar. Within the 100 km radius, the maps are only slightly influenced by the topography and the mean annual rainfall (Journée et al., 2015). This suggests that applying our regional approach is valid, at least for 1 h duration. Van de Vyver (2012) obtained slightly lower values for the 10-year return level but slightly higher 100-year return level due to the positive shape parameter. One notes that the scale is very high around the Brussels region where the Uccle station is located.

## 5 Conclusions

### 5.1 Results

The potential of a radar-based precipitation datasets to study extreme rainfall at a given location is evaluated. The quantitative precipitation estimation (QPE) is obtained by a careful processing of the volumetric reflectivity measurements from a single weather radar in Belgium. The radar dataset covers the period 2005-2016, has a resolution of 1 km, and is available every 5 minutes.

The first evaluation is based on a comparison of the extreme statistics between the radar dataset and two automatic raingauge networks with 10 min and 1 h resolution, respectively. For each network, two locations are chosen to study sliding 1 h and 24 h extremes using the collocated radar estimation. A regression method in Q-Q plots is used to fit an exponential distribution to independent peaks. This method has the property to focus on tail of the extreme value distribution, which is of interest when studying extremes. An optimal threshold rank is selected by minimising the MSE of the regression.

The 10 highest 1 h extremes occurred in summer and are well captured by both the radar and the gauge. A few problematic events are caused by wind drift or severe radar signal attenuation and should be considered as missing data. Differences up to 30 % between the gauge and radar values are observed and can be explained by spatial sampling and estimation errors. The radar extremes tend to be lower than the gauge extremes especially for short return periods. This is consistent with the results of Peleg et al. (2016) on the small scale spatial variability of extreme rainfall. In particular, tipping bucket gauges underestimate heavy rainfall rate and can be blocked by accumulated snow. The radar underestimates due to signal attenuation and overestimates in case of hail. Additional radar uncertainties come from time sampling and the Z-R relationship. Despite the uncertainties in the datasets, the fitting of the exponential distribution to the QPE product is within the large uncertainty bound of the AWS one. This result is in accordance with the fact that the temporal variability (related to the sample size) is higher than the spatial variability (Peleg et al., 2017).

For 24 h accumulation there is a mix of summer and winter events, with more of the latter for stations with higher altitude. There is a clear benefit of bias correction for the highest station, making the distribution fits similar for both stations. For both 1 h and 24 h accumulations, the basic radar product exhibits unrealistic high extremes, which results in an overestimated scale parameter. Such product is therefore not suitable for an extreme value analysis.

In the second evaluation a regional frequency analysis is applied to 1 h radar data at the location of 4 pluviographs with recordings from 1965 to 2010. Spatially independent extremes within a circle of 20 km are selected using a novel approach. They are fitted with a maximum threshold rank extended from 30 to 100 thanks to the increased sample size. There is a good agreement between the radar and the gauge for the two closest stations. The most important result is that the uncertainty is significantly lower using the available radar data. The extremes are lower when a decorrelation distance of 10 km is assumed suggesting that this hypothesis is not valid. In Uccle, the radar extremes and therefore the scale parameter are significantly higher. This can be attributed partially to radar overestimation due to hail and gauge underestimations, but the increasing urban heat island effect should not be ruled out. The decreasing tail of the radar extremes is at least partially caused by hail threshold but a physical limit for the Belgium climate could play a role.

For each of the rain gauge networks, only a few stations have been selected and presented in this paper. The results from these stations are representative of the variability of the results obtained from the other stations.

The regional approach has been applied all over the study area using a 10 km radius and a fixed threshold rank of 60. The extreme statistics for 1 h duration are slightly influenced by the topography. The reliability of the radar results beyond the 100 km range is questionable.

## 5.2 Prospects

There is still some room to improve the quality of the radar and gauge datasets. The recently installed weighting gauges are able to cope with intense rainfall and snowfall. One will have to wait a few decades before it can produce reliable statistics. Radar calibration errors can be mitigated by computing a monthly bias using rain gauges. The attenuation can be solved easily by using other radars when available. To avoid overestimation of the extremes, an advection correction can be used for the time sampling error. Dual-polarization radars can potentially provide better estimation for high rainfall rates (Figueras i Ventura and

Tabary, 2013). However uncertainties related to relation between the radar measurements and the rainfall rate remain, especially in case of hail. In this study we considered all data as the amount of liquid water at the ground. For some applications it could be necessary to take the melting of snow and hail into account. Identification of hail at ground level is a challenging problem using radar and ground station networks (Lukach et al., 2017).

Since the paper focuses on comparison between radar and rain gauges, the extreme value analysis has been kept simple. While the EXP distribution was found to fit generally well with the empirical data, the generalised Pareto distribution should be considered as well for the regional frequency analysis. The analysis of longer durations can be refined by taking into consideration the effect of the type of rainfall (e.g., Rulfova et al., 2014; Panziera et al., 2016). A bias correction should also be considered for a proper handling of the asymptotic behavior of the distribution (Willems et al., 2007).

The extreme value theory was applied to the radar datasets by removing the spatially dependent extremes in the region of analysis. This is performed using a simple technique based on a decorellation distance. Evin et al. (2016) decided not to use such method because it reduces the sample size. Better performance are expected using recently proposed statistical models (Buishand et al., 2008; Davison et al., 2012).

The radar-based regional frequency analysis can be extended to other durations to derive IDF curves. Note that the hypothesis of constant parameter over the region might not be valid for longer durations. In many applications in hydrology, it is the averaged rainfall over a given area which is relevant. A popular technique is to apply areal reduction factors to point-based statistics. The radar dataset can be used directly to derive areal rainfall statistics (e.g., Durrans et al., 2002; Overeem et al., 2010; Wright et al., 2014a).

## 6   Code availability

The code used in this study is part of the RMIB radar library.

## 7   Data availability

The rain gauge rainfall measurements and radar-based precipitation estimates are archived at the RMIB.

*Acknowledgements.* The authors would like to thank Francesco Marra, Luca Panziera and Referee #3 for their very constructive comments which allowed us to significantly improve the quality of the paper. The hydrological service of the Walloon region (SPW) for providing their rain gauge data. The comments of Michel Journee and Hans Van de Vyver are highly appreciated.

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

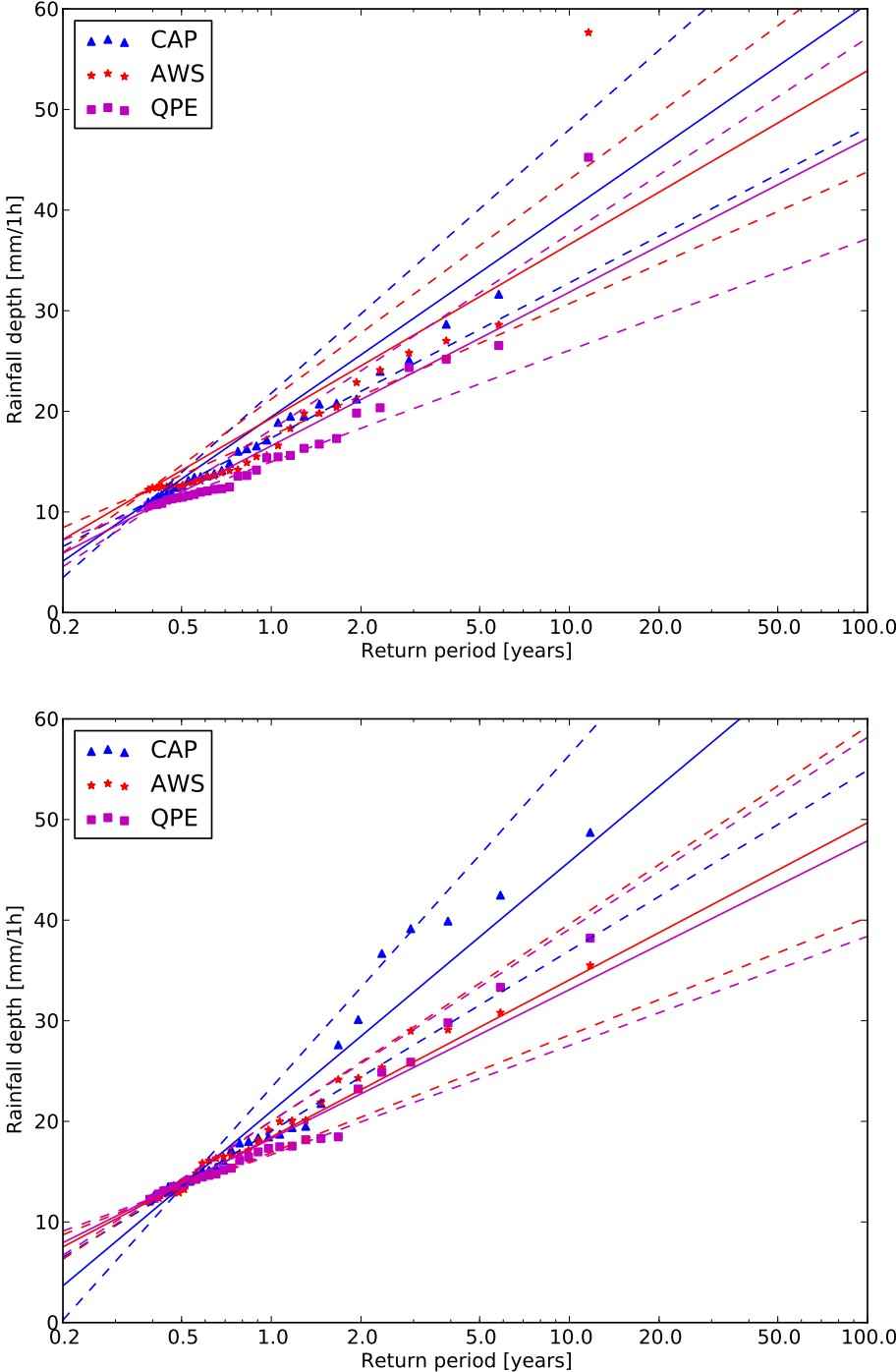

**Figure 2.** Return levels for 1-hour duration at location Humain (top) and Uccle (bottom) of the AWS gauge (red stars) compared to CAP (blue triangles) and QPE (magenta squares) radar products. The extreme value distribution (solid line) fitted to the extremes and its confidence intervals (dashed line) are also displayed.

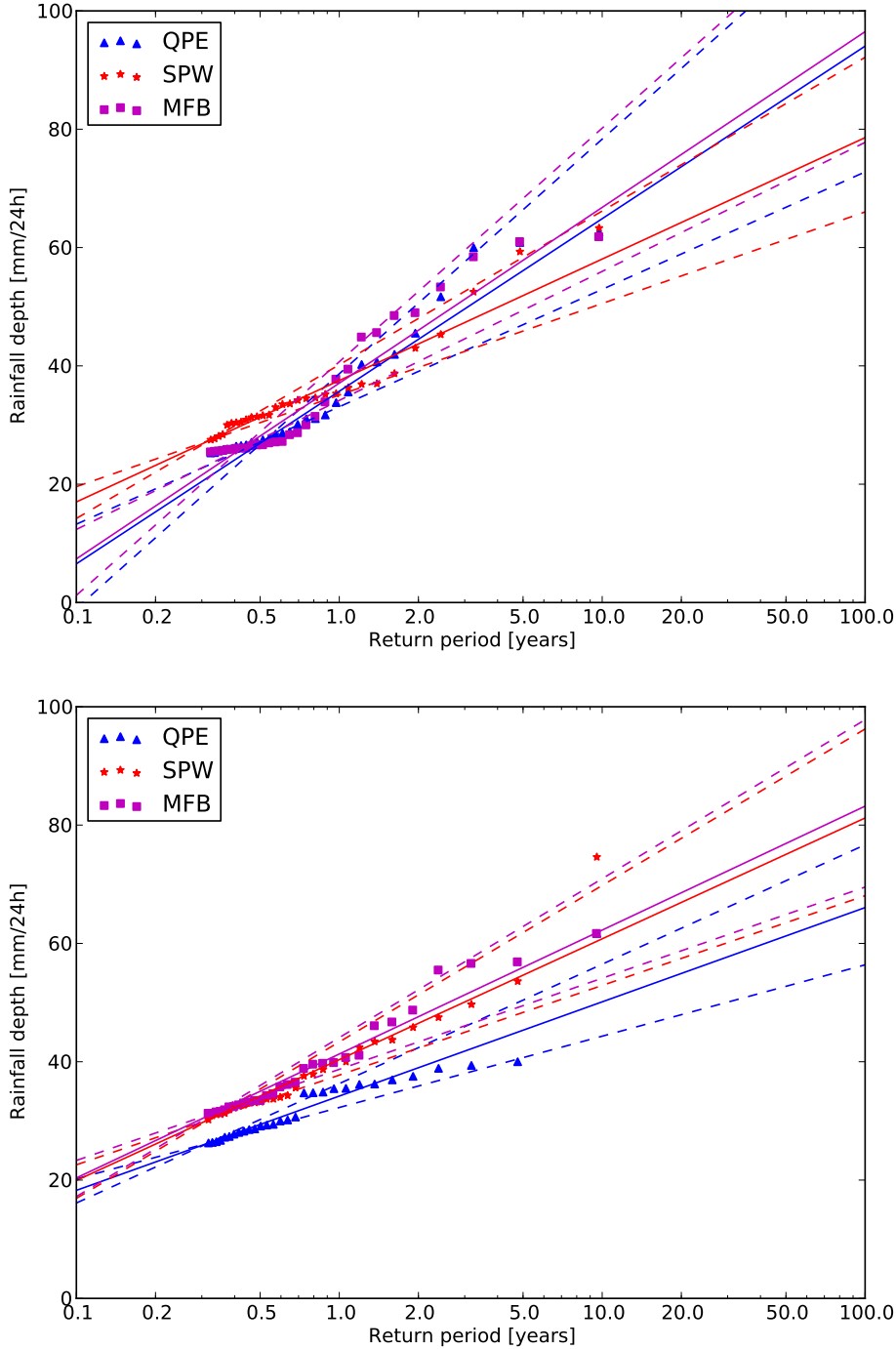

**Figure 3.** Return levels for 24-hour duration at location Uccle (top) and Saint-Vith (bottom) of the SPW gauge (red stars) compared to QPE (blue triangles) and MFB (magenta squares) radar products. The extreme value distribution (solid line) fitted to the extremes and its confidence intervals (dashed line) are also displayed.

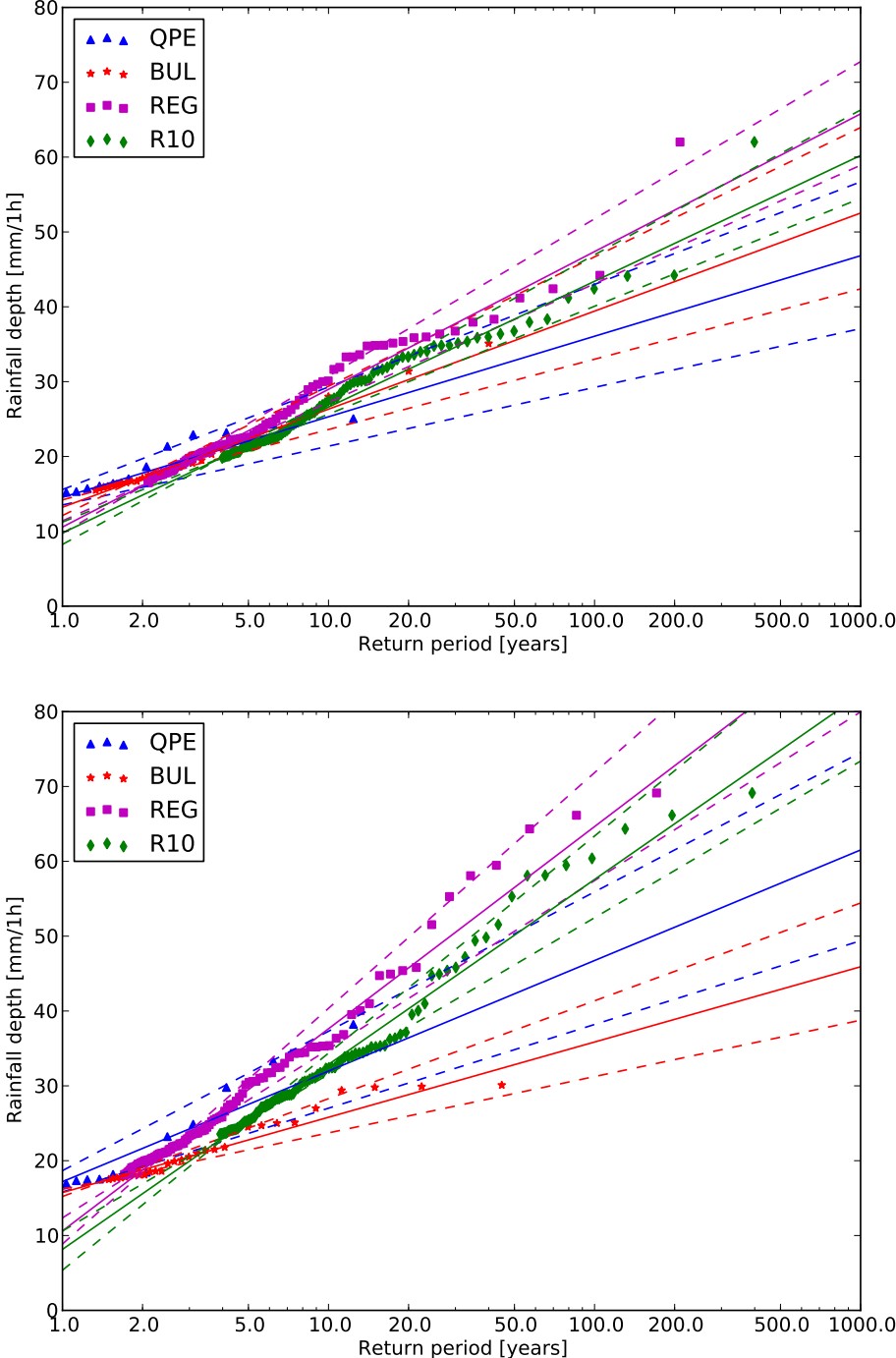

**Figure 4.** Return levels for 1 hour duration at location Deurne (top) and Uccle (bottom) from the BUL gauge data (red stars) compared to the at-site QPE (blue triangle), REG (purple square) and R10 (green diamond) radar data. The extreme value distribution (solid line) fitted to the extremes and its confidence intervals (dashed line) are also displayed.

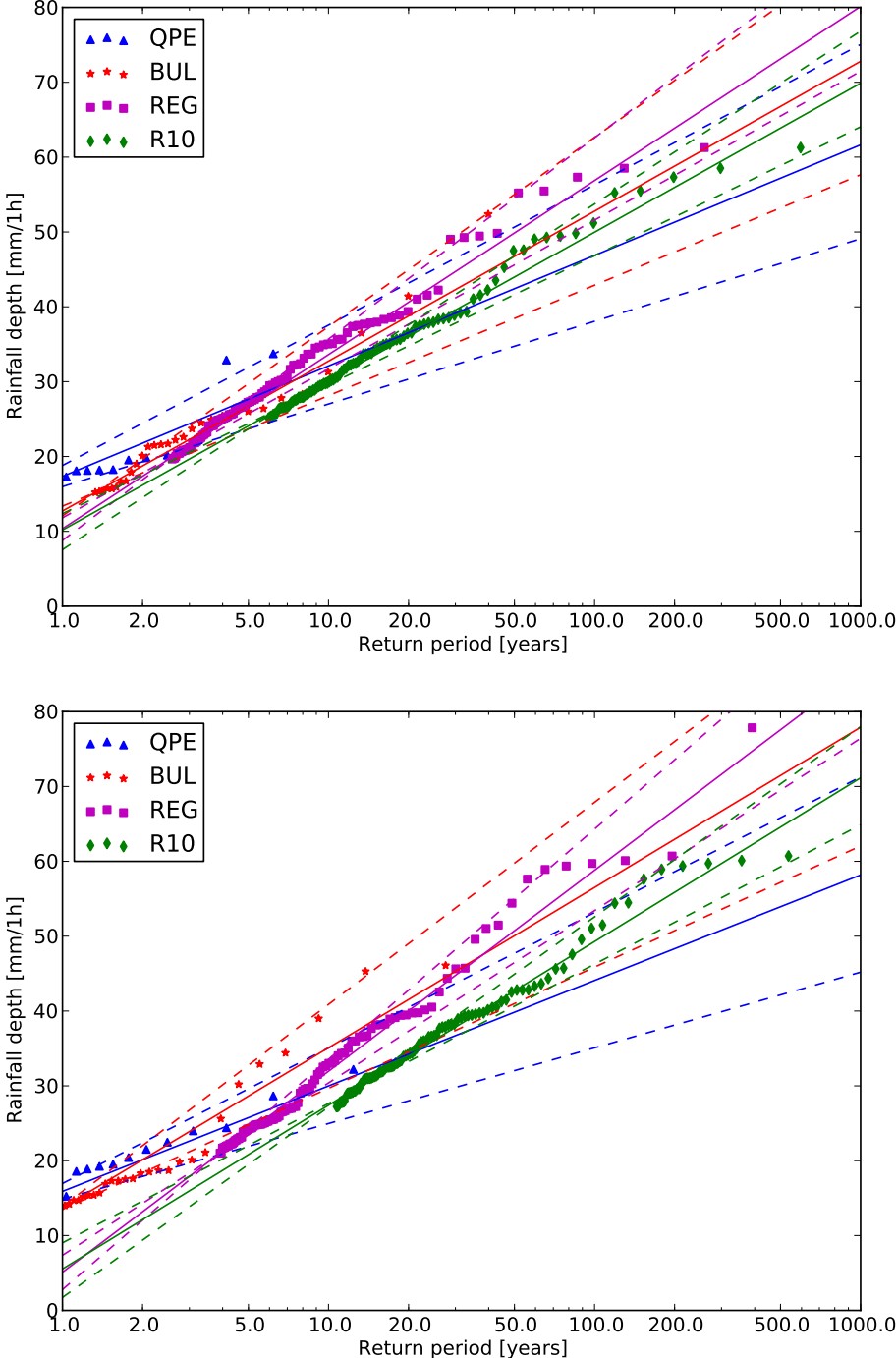

**Figure 5.** Return levels for 1 hour duration at location Gosselies (top) and Nadrin (bottom) from the BUL gauge data (red stars) compared to the QPE (blue triangle), REG (purple square) and R10 (green diamond) radar data. The extreme value distribution (solid line) fitted to the extremes and its confidence intervals (dashed line) are also displayed.

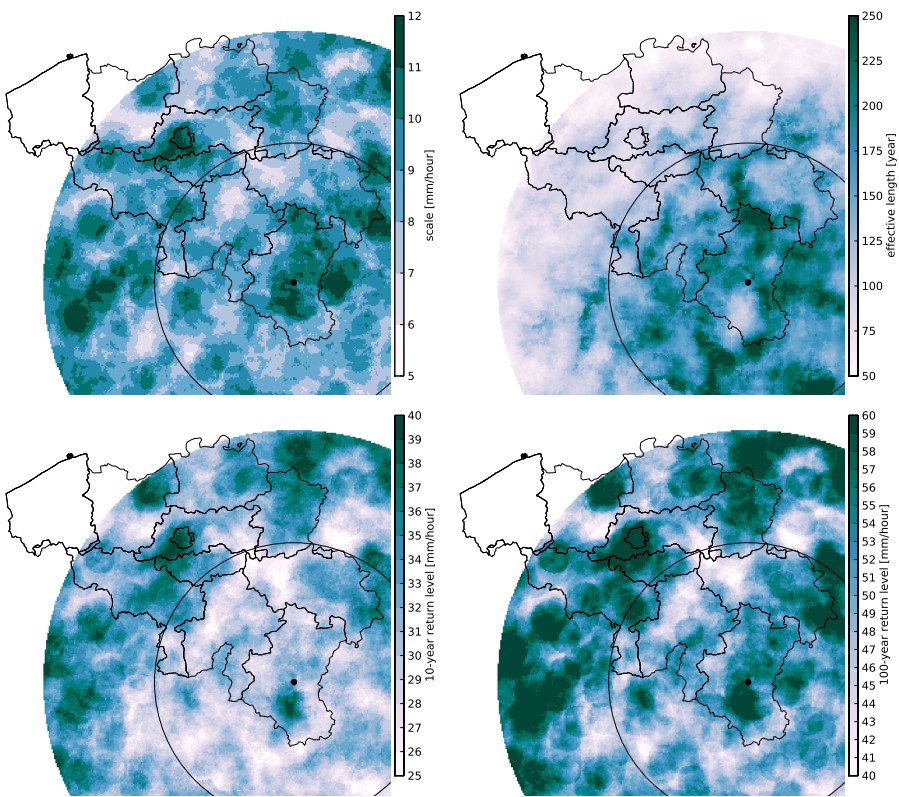

**Figure 6.** Results of the regional frequency analysis for 1 hour duration applied over Belgium up to 180 km from the radar. The scale parameter and the effective length are showned in the top panel. The levels coresponding to a 10-year and 100-year return periods are shown in the bottom panel. A circle with a radius of 100 km centred at the radar is also drawn.

**Table 1.** Rain gauge stations used for comparison and availability of the extreme rainfall datasets. The last column is the percentage of time when both radar and gauge data are available

| Station | Altitude (DNG) | Distance to radar (km) | Duration | Avail. Gauge (%) | Avail. Radar (%) | Avail. Both (%) |
|---------|----------------|------------------------|----------|------------------|------------------|-----------------|
| Humain (AWS) | 296 | 36 | 1h | 98.5 | 94.8 | 93.5 |
| Uccle (AWS) | 100 | 128 | 1h | 99.9 | 94.8 | 94.7 |
| Uccle (SPW) | 100 | 128 | 24h | 90.6 | 86.0 | 78.2 |
| St-Vith (SPW) | 456 | 61 | 24h | 89.2 | 86.0 | 76.7 |
| Deurne (BUL) | 12 | 161 | 1h | 86.0 | – | – |
| Uccle (BUL) | 100 | 128 | 1h | 96.3 | – | – |
| Gosselies (BUL) | 187 | 97 | 1h | 85.7 | – | – |
| Nadrin (BUL) | 403 | 30 | 1h | 59.3 | – | – |

**Table 2.** Comparison of the 10 highest 1-hour rainfall extremes from the gauge (AWS) and radar (QPE) at Humain and Uccle stations. The events with a high probability of hail have their number in bold. The events are ordered by the maximum of the gauge and radar values.

Humain

| Event | Date | Time (end) | Gauge [mm/hour] | Radar[mm/hour] |
|---|---|---|---|---|
| **1** | 2016-06-07 | 18:50:00 | 57.65 | 45.25 |
| 2 | 2005-07-30 | 00:40:00 | 28.60 | 11.62 |
| **3** | 2014-04-24 | 15:40:00 | 27.00 | 20.35 |
| 4 | 2014-06-10 | 21:40:00 | 15.60 | 26.40 |
| 5 | 2007-06-14 | 01:20:00 | 25.80 | 16.32 |
| 6 | 2009-05-25 | 13:10:00 | 24.10 | 25.17 |
| **7** | 2008-05-14 | 17:40:00 | 13.10 | 24.35 |
| 8 | 2015-07-19 | 01:00:00 | 22.87 | 15.47 |
| 9 | 2009-06-27 | 14:30:00 | 20.40 | 19.83 |
| 10 | 2009-07-22 | 21:20:00 | 19.80 | 12.08 |
| 11 | 2010-07-14 | 15:40:00 | 19.80 | —— |
| 12 | 2012-06-12 | 22:20:00 | 18.30 | 15.61 |
| 13 | 2013-03-23 | 07:40:00 | —— | 17.30 |
| **14** | 2005-06-28 | 22:20:00 | —— | 16.74 |

Uccle

| Event | Date | Time (end) | Gauge [mm/hour] | Radar [mm/hour] |
|---|---|---|---|---|
| **1** | 2016-06-07 | 15:20:00 | 18.08 | 38.21 |
| **2** | 2011-08-23 | 08:40:00 | 35.50 | 23.22 |
| **3** | 2009-10-07 | 18:40:00 | 30.79 | 33.32 |
| **4** | 2012-05-20 | 16:30:00 | 12.37 | 29.79 |
| 5 | 2005-09-10 | 19:40:00 | 29.10 | 17.54 |
| **6** | 2011-08-18 | 15:50:00 | 28.98 | 14.77 |
| 7 | 2007-06-14 | 14:50:00 | 21.90 | 25.88 |
| 8 | 2011-09-03 | 22:40:00 | 25.34 | 18.46 |
| 9 | 2016-06-11 | 18:50:00 | —— | 24.88 |
| 10 | 2005-07-29 | 19:10:00 | 24.29 | —— |
| 11 | 2010-07-14 | 15:20:00 | 24.15 | —— |
| 12 | 2014-07-29 | 16:10:00 | 20.10 | 18.17 |
| 13 | 2013-07-27 | 22:20:00 | 20.07 | —— |
| 14 | 2008-07-26 | 10:40:00 | 16.60 | 18.30 |

**Table 3.** Results of the extreme value distribution fitting at two locations of the AWS network. The tables shows successively the temporal independence, optimal rank, the location parameter and the scale parameter. A value is indicated as missing when its extreme rank is below 30

temporal independence [%]

| Station | Gauge | CAP | QPE | MFB |
|---------|-------|------|------|-----|
| Humain  | 25.6  | 20.7 | 22.6 | –   |
| Uccle   | 20.8  | 19.4 | 21.0 | –   |

optimal rank

| Station | Gauge | CAP | QPE | MFB |
|---------|-------|-----|-----|-----|
| Humain  | 30    | 30  | 28  | –   |
| Uccle   | 29    | 23  | 30  | –   |

location parameter [mm/hour]

| Station | Gauge | CAP  | QPE  | MFB |
|---------|-------|------|------|-----|
| Humain  | 12.2  | 11.0 | 10.7 | –   |
| Uccle   | 12.3  | 13.9 | 12.3 | –   |

scale parameter

| Station | Gauge | CAP  | QPE | MFB |
|---------|-------|------|-----|-----|
| Humain  | 7.5   | 8.9  | 6.6 | –   |
| Uccle   | 6.8   | 10.8 | 6.4 | –   |

**Table 4.** Comparison of the 10 highest 24-hour rainfall extremes from the gauge (SPW) and radar (MFB) at Uccle and Saint-Vith stations. A value is indicated as missing when its extreme rank is below 30. The events are ordered by the maximum of the gauge and radar values.

Uccle

| Event | Date | Time (end) | Gauge [mm/24h] | Radar [mm/24h] |
|---|---|---|---|---|
| 1 | 2010-08-16 | 23:00:00 | 63.30 | 48.99 |
| 2 | 2009-10-07 | 23:00:00 | 52.50 | 61.83 |
| 3 | 2011-08-23 | 15:00:00 | 59.31 | 61.00 |
| 4 | 2006-08-03 | 23:00:00 | 43.00 | 58.44 |
| 5 | 2016-05-30 | 23:00:00 | 35.30 | 53.34 |
| 6 | 2014-08-26 | 15:00:00 | 45.30 | 48.51 |
| 7 | 2012-10-04 | 08:00:00 | 34.60 | 45.63 |
| 8 | 2012-06-12 | 11:00:00 | —— | 44.87 |
| 9 | 2016-06-12 | 17:00:00 | 31.30 | 39.45 |
| 10 | 2011-09-04 | 21:00:00 | 38.70 | 26.10 |
| 11 | 2015-08-16 | 03:00:00 | —— | 37.75 |
| 12 | 2007-06-15 | 11:00:00 | 36.99 | 33.91 |
| 13 | 2014-07-10 | 04:00:00 | 36.90 | —— |
| 14 | 2016-01-16 | 02:00:00 | 36.30 | —— |

Saint-Vith

| Event | Date | Time (end) | Gauge [mm/24h] | Radar[mm/24h] |
|---|---|---|---|---|
| 1 | 2007-01-18 | 16:00:00 | 74.60 | 56.88 |
| 2 | 2009-07-03 | 16:00:00 | 37.90 | 61.68 |
| 3 | 2011-12-16 | 23:00:00 | —— | 56.62 |
| 4 | 2012-07-28 | 21:00:00 | 53.60 | 46.72 |
| 5 | 2012-10-04 | 12:00:00 | 49.70 | 39.86 |
| 6 | 2007-08-22 | 19:00:00 | 47.50 | 48.73 |
| 7 | 2010-08-16 | 03:00:00 | 45.80 | 55.50 |
| 8 | 2006-08-05 | 06:00:00 | 43.70 | 41.10 |
| 9 | 2007-12-03 | 08:00:00 | 43.40 | 46.09 |
| 10 | 2007-09-28 | 08:00:00 | 42.40 | 38.87 |
| 11 | 2014-09-21 | 14:00:00 | 34.00 | 40.71 |
| 12 | 2016-05-31 | 02:00:00 | 40.01 | 33.44 |
| 12 | 2016-07-23 | 21:00:00 | 40.00 | —— |

**Table 5.** Results of the extreme value distribution fitting at two locations of the SPW network. The tables shows successively the temporal independence, optimal rank, the location parameter and the scale parameter.

temporal independence [%]

| Station | Gauge | CAP | QPE | MFB |
|---------|-------|-----|-----|-----|
| Uccle   | 7.1   | 6.0 | 6.6 | 6.7 |
| St-Vith | 7.4   | 8.4 | 9.0 | 8.4 |

optimal rank

| Station | Gauge | CAP | QPE | MFB |
|---------|-------|-----|-----|-----|
| Uccle   | 30    | 26  | 19  | 23  |
| St-Vith | 30    | 30  | 30  | 28  |

location parameter [mm/24h]

| Station | Gauge | CAP  | QPE  | MFB  |
|---------|-------|------|------|------|
| Uccle   | 27.2  | 25.0 | 27.2 | 27.5 |
| St-Vith | 30.2  | 25.8 | 26.3 | 31.5 |

scale parameter [mm/24h]

| Station | Gauge | CAP  | QPE  | MFB  |
|---------|-------|------|------|------|
| Uccle   | 9.0   | 13.5 | 12.7 | 12.9 |
| St-Vith | 8.9   | 8.2  | 6.9  | 9.1  |

**Table 6.** Results of the extreme value distribution fitting for the regional frequency analysis. The tables shows successively the independence (temporal or spatial), the optimal rank, the location parameter and the scale parameter.

independence [%]

| Station | QPE | BUL | R50 | R10 |
|---|---|---|---|---|
| Deurne | – | 27.5 | 1.4 | 2.6 |
| Uccle | – | 28.0 | 1.1 | 2.6 |
| Gosselies | – | 22.2 | 1.7 | 3.9 |
| Nadrin | – | 19.9 | 2.6 | 7.0 |

optimal rank [%]

| Station | QPE | BUL | R50 | R10 |
|---|---|---|---|---|
| Deurne | 28 | 22 | 100 | 99 |
| Uccle | 30 | 30 | 70 | 88 |
| Gosselies | 29 | 30 | 96 | 90 |
| Nadrin | 23 | 30 | 100 | 91 |

location parameter [mm/hour]

| Station | QPE | BUL | R50 | R10 |
|---|---|---|---|---|
| Deurne | 10.8 | 16.7 | 16.5 | 20.0 |
| Uccle | 11.5 | 17.5 | 21.1 | 24.2 |
| Gosselies | 11.9 | 15.2 | 20.4 | 26.5 |
| Nadrin | 12.2 | 12.9 | 21.0 | 29.0 |

scale parameter [mm/hour]

| Station | QPE | BUL | R50 | R10 |
|---|---|---|---|---|
| Deurne | 4.7 | 5.7 | 8.0 | 7.3 |
| Uccle | 6.4 | 4.4 | 11.7 | 10.7 |
| Gosselies | 6.4 | 8.7 | 10.1 | 8.6 |
| Nadrin | 6.1 | 9.3 | 11.7 | 9.5 |