# Peer review of "At-site and regional frequency analysis of extreme rainfall in Belgium based on radar estimates"

_Hydrology and Earth System Sciences, 2017_

## Referee Comment (RC1) · F. Marra (Referee) · 21 May 2017

**hess-2017-150 – At-site and regional frequency analysis of extreme precipitation from radar-based estimates – by Goudenhoofdt et al**

The manuscript presents the use of different methods to perform rainfall frequency analysis from weather radar data. The topic is of increasing interest for the community given the growth of radar and remote sensing archives worldwide. Studies proposing methods, testing approaches and evaluating the accuracy of such products are greatly welcome and definitely of interest for the readers of HESS.

This study focuses on a region in Belgium and derives at-site and 'regional' frequency analyses for 1h and 24h durations and provides new contributions (not clearly evident from the text) to the field, such as the use of new (i) methods (i.e. a peak over threshold approach, QQ plot regression) and (ii) regionalization approaches for rainfall frequency analyses from remote sensing data sets.

This study will provide new, interesting information to the field and deserves publication. However, a number of issues are currently preventing it from being published in its present form: literature review is missing key papers that need to be mentioned and, in some cases, discussed; methods are not sufficiently described, motivated and supported by literature; some of the results need to be re-considered and discussed, also in light of the new literature review; presentation and language need some improvement. Below a list of major and minor comments.

**Major comments**

1. **Literature review** is missing some key papers of the field.
   - Panziera et al., 2016 developed regional rainfall frequency analysis and implemented (and tested) them in an early-warning system for Switzerland – this is probably the first study deriving rainfall frequency analysis from remote sensing data and providing an actual operational, quantitative product;
   - Peleg et al., 2017 analyzed the impact of small scale rainfall variability on frequency analysis from point (rain gauges) and areal (radar) estimates – they found that, due to the relatively short record length, point and areal estimates, are expected to differ (even if both measurements are 'true'), and observed large differences between frequency analyses from rain gauges located within a 1 km$^2$ pixel. This means that no exact match between point and areal results should be expected (not only because of the areal reduction factors issue). This contribution needs to be mentioned by the literature review, in light of the contribution it provides to the interpretation of the results of this study (also the conclusions [page 11, lines 18-19] could be updated accordingly);
   - Wright et al., 2013 proposed the use of stochastic storm transposition for radar rainfall frequency analysis in order to overcome the limitations due to the short radar records.

2. **Methods** are sometimes insufficiently described and apparently subjective choices are made without providing the reader with rationale, supporting references, and discussion of the implications.
   *Frequency analysis:*
   - the use of PoT approach for the frequency analysis of short records is highly desirable, however the choice of the exponential distribution (a special case in which the shape parameter – driving the long return period tail of the distribution – is assumed uniform in space and equal to 0) is very strong and goes against some literature on the topic. This choice needs to be motivated and supported;
   - [page 6, lines 13-18] how is the return period used in the QQ-plots derived? Is it is done following Willems et al, 2007? This is a key aspect in the methods and in the shaping of the results and needs to be explained and discussed; [7, 12] how is figure 2 created? from Willems et al, 2007, I imagine that the figure shows on the x-axis the return period derived from the exponential distribution that maximizes the linearity of the relation – how is the maximization done?;
   - [6, 14-16] this is not clear for the reader unfamiliar with Willems et at., 2007, please provide more information;

- [7, 20] what do the authors mean with "heavy tailed" (the exponential distribution has no shape parameter)?
- [9, 7-8] this sentence should be somehow supported/motivated;
- [10, 14] why is 60 chosen?

*Radar QPE:*
- [9, 22-23], what do the authors mean with "standard Z-R"? In what cases is a non-standard Z-R used?
- Previous studies found important instability of the MFB factor for short periods (1 h), especially in convective conditions. The use of hourly mean field bias adjustment needs to be supported by sensitivity analyses or references. Please discuss this and provide information on the stability of the factors;
- Is there any motivation for the choice of the hail threshold (80 mm/h looks low for some climates)? Are there cases in which the rain gauges did measure heavier rain intensities? [7, 18] is it possible to check how often the hail filter is activated? Is it possible to check what reported in [7, 23-24]?
- [8, 5] is it possible to check if bright band was contributing to this observation? This would be interesting since VPR impact was rarely discussed in previous studies on radar-based frequency analyses;

3. **Interpretation** of some results.
- [Section 4.1] The authors select the maximum within 20 km range windows around each analyzed pixel in order to better capture the maximal intensities (see [8, 31]). The motivation for doing this is clear to me and it is a good direction to take to exploit the distributed information provided by the radar (and other gridded datasets). However, I am wondering whether the interpretation one should give to the obtained results still holds: will we still be dealing with the estimation of the frequency of occurrence of a given rain intensity-duration combination at a given location? Especially since the conclusions open with: "... to study extreme precipitation at a given location...". I'm not sure this is the case. I recommend the authors carefully examine and discuss this issue. At this regard, the stochastic storm transposition approach adopted by Wright et al., 2013, even though much more complicated, provides similar advantages while preserving the interpretation;
- [Figure 6, 7] Can the circular patterns be caused by the regionalization method (in case problematic pixels are still there, one will choose them when selecting the max value within the circular area – see also [10, 20-21])? Can this represent a weakness of this method?

4. **Presentation** is sometimes difficult to follow.
- new contributions brought by this study are not clearly stated in abstract, introduction and conclusions. Reading the manuscript, the main results appear to be: raw radar QPE provides unreliable analyses and bias adjustment is needed; differences are observed between at-site analyses from radar and gauges, but radar analyses lie within the gauge confidence intervals; regionalization approaches provide improved analyses. These results were already reported in literature (see for example Overeem et al, 2009; Marra and Morin, 2015; Panziera et al, 2016; Peleg et al 2017; Marra et al., 2017). In my opinion the study brings a lot of new to the field, in particular the use of (i) new methods (i.e. a peak over threshold approach, QQ plot regression) and (ii) new regionalization approaches for rainfall frequency analyses from remote sensing data sets. Abstract, introduction and conclusions need to be reorganized in these terms, even though the results reported by the authors definitely deserve to be mentioned;
- the presentation of the gauge networks in [2, 20-25] and in section 2.1 is difficult to follow, I recommend reorganizing and rephrasing these parts (how many networks are used?, why are they considered separately?, what are the differences? What the advantages of including each of them? Why not using them together?);
- [4, 8-9] please provide information about these methods and move the reference to Goudenhoofdt and Delobbe, 2016 earlier in the text;
- organization of the radar datasets (QPE, CAP,...) need to be made clearer (sections 2.2, 2.3);
- [section 3.1] is difficult to follow; in particular [6, 4-9] and [6, 19-21] are not clear to me;
- [section 3.2] what does "problematic events" mean? How are they identified?; why did the authors focus on 10 extremes?
- [8, 21-22] why mentioning the index flood approach? Here the shape is actually assumed uniform (by the use of

the exponential distribution), but I guess this is not what the authors mean with 'regionalization';

- how does the 20 km regionalization of the parameters relate to the 10 km and 50 km used in the following parts of the study? how did the authors check/motivate that 20 km "provides a sufficiently large data sample"?
- it is not clear whether the method by Reed et al., 1999 is the one the authors used in this paper;
- [9, 28-29] did the authors check for non-stationarity in the data (e.g. changes in the instrumentation, or other)?
- [11, 14-15] is this expected?
- [11, 24-25] this problem should be solved by the adopted regionalization (20 km);
- [1, 11-13], [4, 8], [4, 20-24], [6, 19-21], [7, 10-11], [9, 1-5], [9, 11-16], [9, 31-32] please rephrase;

**Minor comments/edits**

- [abstract] the text of the abstract needs to be better organized;
- [page 1, line 3] "independent sliding 1h and 24h rainfall": this is not clear;
- [1, 9] natural rainfall variability in combination with short record lengths is also to be mentioned as a cause of the mismatch between point and areal frequency analyses (see Peleg et al., 2017 and major comment above);
- [1, 11] "assuming that the extremes are correlated": this is not clear to me, I guess it is related to the regionalization, but needs to be better written;
- [1, 18] please remove "very" and "very"; please add a comma after "activities"; please provide a reference for this sentence;
- [1, 19] sewer systems are an example, but I'd insert an example from other applications, such as dams design/management; [1, 21] sewer systems are usually designed for relatively short return periods, applications requiring long return periods are dams, bridges, etc.;
- [1, 20] no need to specify "a branch of statistics";
- [2, 3] an example of what the authors mean with "high-resolution" would be helpful for the reader;
- [2, 5-6] "the best potential is provided by radar QPE". Satellite products are fruitfully being used too and are, often, characterized by longer records. This sentence should be motivated and supported by references;
- [2, 11-12] the reference to Saito and Matsuyama, 2015 looks unrelated to the rest of the text, can you provide some information on its relevant parts;
- [Figure 1] what do the areas in the figure represent? Are they catchments? Are they used in the manuscript?
- [4, 28] how is gauge data validated?
- [5, 5-6] "The hourly bias obtained… convective storms" can be removed;
- [5, 6-10] is this done with a moving window? Or on 24 h blocks?
- [5, 10-11] Marra and Morin, 2015 quantified this uncertainty;
- [Table 1] what is the meaning of the "Avail. All" column? Does it mean that "Both" were available?
- [6, 22 and 7, 25] I'd suggest to change these titles to something focusing on the tested product rather than on the rain gauges against which it is compared;
- [8, 21-22] please change "consider" to "considers";
- [10, 25] please, change "estimate" to "estimation";
- [11, 18-19] the authors may want to check Avanzi et al., 2015 for additional inputs;
- [12, 6] since the work by Frederick et al., 1977, a number of papers are available on the derivation of ARFs from radar data (e.g. Durrans et al., 2002; Overeem et al., 2010; Wright et al., 2014, among others).

**References**

Avanzi, F. et al. 2015. "Orographic Signature on Extreme Precipitation of Short Durations." Journal of Hydrometeorology 16(1): 278–94. http://journals.ametsoc.org/doi/abs/10.1175/JHM-D-14-0063.1.

Durrans, S. Rocky, Lesley T. Julian, and Michael Yekta. 2002. "Estimation of Depth-Area Relationships Using Radar-Rainfall Data." Journal of Hydrologic Engineering 7(5): 356–67.

Frederick, Ralph H., Vance A. Myers, and Eugene P. Auciello. 1977. "Storm Depth-area Relations from Digitized Radar Returns." Water Resources Research 13(3): 675–79.

Goudenhoofdt, E. and Delobbe, L.: Generation and Verification of Rainfall Estimates from 10-Yr Volumetric Weather Radar Measurements, Journal of Hydrometeorology, 17, 1223–1242, doi:10.1175/JHM-D-15-0166.1, http://dx.doi.org/10.1175/JHM-D-15-0166.1, 2016

Marra, F. and E. Morin 2015: Use of radar QPE for the derivation of Intensity–Duration–Frequency curves in a range of climatic regimes. Journal of Hydrology 531 (2015) 427–440. http://dx.doi.org/10.1016/j.jhydrol.2015.08.064

Marra, F., Morin, E., Peleg, N., Mei, Y., and Anagnostou, E. N.: Intensity–duration–frequency curves from remote sensing rainfall estimates: comparing satellite and weather radar over the eastern Mediterranean, Hydrol. Earth Syst. Sci., 21, 2389-2404, doi:10.5194/hess-21-2389-2017, 2017

Overeem, A., T. A. Buishand, and I. Holleman. 2009. "Extreme Rainfall Analysis and Estimation of Depth-Duration-Frequency Curves Using Weather Radar." Water Resources Research 45(10): 1–15.

Overeem, A., T. A. Buishand, I. Holleman, and R. Uijlenhoet. 2010. "Extreme Value Modeling of Areal Rainfall from Weather Radar." Water Resources Research 46(9): 1–10.

Panziera, L., Gabella, M., Zanini, S., Hering, A., Germann, U., and Berne, A.: A radar-based regional extreme rainfall analysis to derive the thresholds for a novel automatic alert system in Switzerland, Hydrol. Earth Syst. Sci., 20, 2317–2332, doi:10.5194/hess-20-2317-2016, 2016.

Peleg, N., Marra, F., Fatichi, S., Paschalis, A., Molnar, P., and Burlando, P.: Spatial variability of rainfall at radar subpixel scale, J. Hydrol., in press, doi:10.1016/j.jhydrol.2016.05.033, 2017.

Reed, D. W., Faulkner, D. S., and Stewart, E. J.: The FORGEX method of rainfall growth estimation II: Description, Hydrology and Earth System Sciences, 3, 197–203, doi:10.5194/hess-3-197-1999, http://www.hydrol-earth-syst-sci.net/3/197/1999/, 1999.

Willems, P., Guillou, A., and Beirlant, J.: Bias correction in hydrologic GPD based extreme value analysis by means of a slowly varying function, Journal of Hydrology, 338, 221, doi:10.1016/j.jhydrol.2007.02.035, 2007.

Wright, Daniel B., James A. Smith, Gabriele Villarini, and Mary Lynn Baeck. 2013. "Estimating the Frequency of Extreme Rainfall Using Weather Radar and Stochastic Storm Transposition." Journal of Hydrology 488: 150–65. http://dx.doi.org/10.1016/j.jhydrol.2013.03.003.

Wright, Daniel B., James A. Smith, and Mary Lynn Baeck. 2014. "Critical Examination of Area Reduction Factors." Journal of Hydrologic Engineering 19(4): 769–76.

---

## Referee Comment (RC2) · L. Panziera (Referee) · 9 Jun 2017

*Hydrol. Earth Syst. Sci. Discuss.*

**At-site and regional frequency analysis of extreme precipitation from radar-based estimates**

E. Goudenhoofdt, L. Delobbe, P. Willems

In this paper a peak-over-threshold method is used to perform an extreme rainfall analysis and to derive return levels from weather radar and rain gauges in Belgium. The importance of this work is high, as radar archives are nowadays long enough to permit the development of extreme rainfall analyses which are of fundamental importance for many applications, but the common annual maxima approach needs even longer time series. However, some important explanations and discussions, in addition to those already highlighted by the first review by F. Marra, are missing in the manuscript, and need to be provided before the article can be accepted for publication.

Luca Panziera

**Major comments**

1. **What is new?** It is somehow difficult to understand which new contribution this paper brings with respect to previous studies, and I think that this should be better highlighted in the text. To my understanding, the main novelty of this paper is the use of a POT method for an extreme rainfall analysis for weather radar data.

2. **Temporal Declustering**. As rainfall data need to be declustered in order to remove the temporal correlation in the time series before GPD parameters estimation, the authors choose an interval of 12 hours (for 1-hour rainfall) and 3 days (for daily rainfall) in order for two threshold exceedances to be considered as independent. The choice of these intervals, which should be referred to as run length or run parameters according to the literature, seems reasonable, but it could potentially have a big impact on the derived return levels, as it shapes the exceedances time series whose maxima are used for the parameters estimation. If the data are temporally clustered, such temporal lags could not be long enough to remove dependency, but if the temporal clustering occurs rarely, they could actually lead to a significant bias of the return levels estimates. What do the authors mean as temporal independence? How did the authors choose such temporal lags? Did the authors investigate the effect of changing these values on the parameters estimation and final return levels? The subjective choice of these values should be motivated and discussed in the text.

3. **Exponential distribution**. As the choice of a null shape parameter is fundamental for this work, I think that it should be motivated more in the text. Therefore, I suggest to briefly report and discuss the main results of Willems (2000), in order to better understand the motivation of this choice The text states also that this choice was taken because of the short period. However, with a POT approach the shortness of the period should not be a limiting factor, as many events are considered. It should also be discussed if this is the best choice for both 1-hour and 24-hours accumulations. Did the authors try to estimate also the shape parameters, to see if from the data a value different from 0 could be derived?

4. **Radar and gauge comparison.** The authors present an interesting comparison between the radar and gauges extremes, for 1-hour and 24-hours accumulations. Despite this being very interesting and

instructive, the implications for this study are not very clear. I suggest the authors discuss at least qualitatively the influence of this investigations on the overall results of the study.

5. **Regional frequency analysis**. The regional frequency analysis needs also to be better explained and the choices which were taken need to be motivated and discussed. How did the authors choose the 20-km radius for the analysis? How the resulting return levels at a given pixel should be interpreted, as they stem from exceedances in rainfall values which were observed all around it? Does it still make sense to speak about point measurement? How are the maps of GPD parameters affected by the choice of the 20-km radius circles?

6. **Return levels maps**. I guess the final goal of the study is to derive maps of return levels with relative uncertainty for Belgium. Despite the return levels are shown for given rain gauge locations, it would be desirable to show also maps of return levels for selected return periods. Would it be possible to insert a map or two of the return levels? How would those maps be affected by the 20-km radius selected for the regional frequency analysis? How these maps should be interpreted? Since you are using a constant shape parameter (equal to 0), and the longest return levels are shaped by it, long return periods map will tend to produce maps less variable in space with respect to short return periods. This should be discussed in the text.

**Minor comments**

1. The title is rater general, and you might want to consider adding the name of the region for which this study was performed (Belgium).
2. In the introduction some relevant papers are missing. I strongly encourage the authors to discuss also the papers referenced by F. Marra in his review.
3. Pag.2, line22."*in this study, we want to demonstrate the potential of this radar-based QPE to derive point rainfall statistics*". I don't think that the aim of this study is this, as the radar pixel does not represent point rainfall statistics. As the authors know, the radar rainfall estimate comes from the reflectivity measured within the sample volume, representing an area- not a point- measurement. The intrinsic difference among radar and gauges measurements should be at discussed in the paper, since a comparison between rain gauges and radar return levels is performed (see also major comment 1 by F. Marra).
4. Pag.3, line 4: is there a reference for the 5-10% rain gauges underestimation?
5. Pag.3, line 7: improve English. I propose to change "very high" with "10-min" temporal resolution (and delete *"10-min accumulations are available from the database"*)
6. Pag. 4, line 25: please clarify the last sentence of section 2.2 which, in its present form, it is not correct. Could change *" In addition, the increasing radar sample volume will give lower extreme values"* to *"In addition, the increasing radar sample volume will produce an underestimation of local small-scale extremes"*.
7. Pag. 5, line 24. First two sentences of section 3.1 need to be reformulated as they are very colloquial.
8. Page 6, line 14. With this method of regression in QQ plots, is there a risk of over fitting? Could you please comment on that?
9. pag.7, line 13-14 and pag.8 lines 9-10. How this percentage would vary by changing the temporal lags considered for independence? (see major comment 2). *"This is what we expect from …."*. According to which theory/observations? Please clarify and give references.
10. Pag. 8, lines 21-28. It would be more appropriate move the literature review to the Introduction, instead of leaving it in this Methodology section.

11. Pag. 8, second paragraph of section 4.1: please clarify the explanation of the regional frequency analysis. Given that your circle has a radius of 20 km, what is the aim of considering all the events within a 50 km radius dependent? Isn't this the same as taking just the max value within the 20-km radius? In case it is, wouldn't be easier just say that you take this maximum within the 20-km radius circle?

12. Pag.10, lines 6-9. Also here I suggest to move the references to other studies in the Introduction.

13. Pag. 10, line 13: *"a few pixels having too much (50) …. removed"* . This sentence is rather unclear, and this seems a rather subjective choice which can hardly be motivated.

14. Figure 2. I suggest to rename "Extreme 1-hour precipitation quantiles" to "1-hour return levels", to be consistent with theory and common nomenclature in the field.

15. Tables 2 and 4. I actually miss how the events in the tables are ordered, if there is a logic.

16. Figures 1, 6 and 7: a scale in km would help the interpretation of the figure, for those who are not familiar with Belgium

---

## Referee Comment (RC3) · Anonymous Referee #3 · 12 Jun 2017

General comments:

The authors apply local and regional frequency analysis (RFA) for extreme rainfall on two radar data products (advanced QPE and basic CAP) for Belgium and compare the results with station based extreme value statistics. They find that the basic radar product shows unrealistic high extremes, the 24h extremes need bias correction and that the fit of the QPE probability distribution is within the confidence interval of the point distribution. The results for RFA are more complex.

The topic of the paper is very important and of high relevance for the community. The results are interesting. However, the description of methodology is not clear enough

to follow the procedures and understand all the results. This concerns especially the sampling strategy for RFA. Also the presentation of results could be more distinct. Details are given below. However, the research is worth of publication after the authors have the opportunity to make some revisions.

Detailed comments:

1. Abstract, lines 10-15: I cannot really understand these sentences: RFA within 20 km?, which region(s)?, rain gauge vs. automatic gauge?, which radar product?, etc.

2. Page 6, lines 23ff: It is not fully clear if the 10 highest gauge extremes or the 10 highest radar extremes are selected. In the abstract "rain gauges and collocated radar estimates" is mentioned, so I assume the highest gauge values with collocated radar data are used. This should be stated clearly here in the text as well. The rational for this choice should also be discussed.

3. Page 7, lines 26ff: see comment 2

4. Page 8, section 4.1: The sampling for RFA is not clear to me. Do you do a separate RFA for each 20 km radius? How can you apply a minimum distance of 50 km to secure independence with a 20 km radius? If you apply RFA for each radar pixel and consider a minimum distance of e.g. 10 km, then the (collocated) sample is different for each estimate? What about the "index rainfall"? How did you regionalise it? etc.

5. Page 10, lines 11-12: ".. using a radius of 10 km (with a decorrelation distance of 50 km)" I don't understand this.

6. Fig. 2-5: The many lines in in these figures are hardly to disentangle visually. I have not really a good idea what to do here, may be showing only two distributions with confidence limits or excluding the confidence limits of the radar data, or showing additionally bar plots with a comparison of selected quantiles, etc.?
* * *
150, 2017.

---

## Author Comment (AC1) · 20 Jun 2017

**Authors response to Referee 1 comments**

The manuscript presents the use of different methods to perform rainfall frequency analysis from weather radar data. The topic is of increasing interest for the community given the growth of radar and remote sensing archives worldwide. Studies proposing methods, testing approaches and evaluating the accuracy of such products are greatly welcome and definitely of interest for the readers of HESS.

This study focuses on a region in Belgium and derives at-site and 'regional' frequency analyses for 1h and 24h durations and provides new contributions (not clearly evident from the text) to the field, such as the use of new (i) methods (i.e. a peak over threshold approach, QQ plot regression) and (ii) regionalization approaches for rainfall frequency analyses from remote sensing data sets.

This study will provide new, interesting information to the field and deserves publication. However, a number of issues are currently preventing it from being published in its present form: literature review is missing key papers that need to be mentioned and, in some cases, discussed; methods are not sufficiently described, motivated and supported by literature; some of the results need to be re-considered and discussed, also in light of the new literature review; presentation and language need some improvement. Below a list of major and minor comments.

We would like to thank Francesco Marra for summarising the value of the paper and his in-depth analysis of our work. It allowed us to improve the paper and to put some results in perspective.

**Major comments**

**Literature review**

Literature review is missing some key papers of the field.

- Panziera et al., 2016 developed regional rainfall frequency analysis and implemented (and tested) them in an early-warning system for Switzerland – this is probably the first study deriving rainfall frequency analysis from remote sensing data and providing an actual operational, quantitative product;

This paper focusing on areal maximum extremes has been added in the introduction. Please see page 3, line 25.

- Peleg et al., 2017 analyzed the impact of small scale rainfall variability on frequency analysis from point (rain gauges) and areal (radar) estimates – they found that, due to the relatively short record length, point and areal estimates, are expected to differ (even if both measurements are 'true'), and observed large differences between frequency analyses from rain gauges located within a 1 km 2 pixel. This means that no exact match between point and areal results should be expected (not only because of the areal reduction factors issue). This contribution needs to be mentioned by the literature review, in light of the contribution it provides to the interpretation of the results of this study (also the conclusions [page

11, lines 18-19] could be updated accordingly);

The sub-pixel rainfall spatial variability explains partly the different frequencies. This reference and a related paper are now used in the text. Please see page 3, line 5 ; page 14, line 1 and page 14, line 7. I understand that there are spatial dependence within the pixel and that for higher return periods the higher 'climatic' variability is dominant. It is unclear however if one can extrapolate this result for a larger region. In our study we make the assumption that the 1 hour extremes occurring in the 20 km region on different days are independent.

- Wright et al., 2013 proposed the use of stochastic storm transposition for radar rainfall frequency analysis in order to overcome the limitations due to the short radar records.

This study focusing on catchment-averaged precipitation contains very interesting results for single radar pixel. Their methodology is very similar to our work so it is now referenced in the text. Please see page 3, line 21 ; page 11, line 5 ; page 11, line 10 and page 11, line 22.

**Methods**

Methods are sometimes insufficiently described and apparently subjective choices are made without providing the reader with rationale, supporting references, and discussion of the implications.

**Frequency analysis**

- the use of PoT approach for the frequency analysis of short records is highly desirable, however the choice of the exponential distribution (a special case in which the shape parameter – driving the long return period tail of the distribution – is assumed uniform in space and equal to 0) is very strong and goes against some literature on the topic. This choice needs to be motivated and supported;

Please see from page 8, line 3.

- page 6, lines 13-18: how is the return period used in the QQ-plots derived? Is it is done following Willems et al, 2007? This is a key aspect in the methods and in the shaping of the results and needs to be explained and discussed; [7, 12] how is figure 2 created? from Willems et al, 2007, I imagine that the figure shows on the x-axis the return period derived from the exponential distribution that maximizes the linearity of the relation – how is the maximization done?;

It is unclear what the referee means by the "usage" of return periods in QQ-plots. The text related to Figure 2 has been improved. Please see page 9, line 19. The optimisation procedure is explained on page 8, line 20.

- This is not clear for the reader unfamiliar with Willems et at., 2007, please provide more information;

The QQR method is now briefly described in the text. Please see from page 8, line 10.

– 7, 20: what do the authors mean with "heavy tailed" (the exponential distribution has no shape parameter)?

The comment refers to the empirical quantiles and has therefore been moved to the corresponding paragraph. Please see page 9, line 28.

– 9, 7-8: this sentence should be somehow supported/motivated;

This is a consequence of the previous sentences. The text has been adapted on page 11, line 26.

– 10, 14: why is 60 chosen?

Please see page 13, line 3.

**Radar QPE**

– 9, 22-23: what do the authors mean with "standard Z-R"? In what cases is a non-standard Z-R used?

The text has been clarified. Please see page 12, line 10.

– Previous studies found important instability of the MFB factor for short periods (1 h), especially in convective conditions. The use of hourly mean field bias adjustment needs to be supported by sensitivity analyses or references. Please discuss this and provide information on the stability of the factors;

As stated, the bias corrected hourly accumulations are only used to study 24 hours precipitation extremes. We added some discussions on page 6, line 34 and page 10, line 30.

– Is there any motivation for the choice of the hail threshold (80 mm/h looks low for some climates)? Are there cases in which the rain gauges did measure heavier rain intensities? [7, 18] is it possible to check how often the hail filter is activated? Is it possible to check what reported in [7, 23-24]?

The choice of the hail threshold is more discussed starting from page 6, line 8. The probable hail events are now indicated and discussed starting from page 8, line 28.

– 8, 5: is it possible to check if bright band was contributing to this observation? This would be interesting since VPR impact was rarely discussed in previous studies on radar-based frequency analyses;

It appears that it is the bias correction which contributes the most to the overestimation. Please see page 10, line 15.

**Interpretation of some results**

– Section 4.1: The authors select the maximum within 20 km range windows around each analyzed pixel in order to better capture the maximal intensities (see [8, 31]). The motivation for doing this is clear to me and it is a good direction to take to exploit the distributed information provided by the radar (and other gridded datasets). However, I am wondering whether the interpretation one should give to the obtained results still holds: will we still be dealing with the estimation of the frequency of occurrence of a given rain intensity-duration combination at a given location? Especially since the

conclusions open with: "... to study extreme precipitation at a given location...". I'm not sure this is the case. I recommend the authors carefully examine and discuss this issue. At this regard, the stochastic storm transposition approach adopted by Wright et al., 2013, even though much more complicated, provides similar advantages while preserving the interpretation;

We are not interested in the regional maximum extreme but in the extreme at a given location, hence the comparison against gauge data. If we understood correctly, Wright et al. (2014b) (section 3.2) also take the maximum values in the region within a 24 h window. This is now better discussed from page 11, line 10.

– Figure 6, 7: Can the circular patterns be caused by the regionalization method (in case problematic pixels are still there, one will choose them when selecting the max value within the circular area – see also [10, 20-21])? Can this represent a weakness of this method?

This is now further discussed in the text. Please see from page 13, line 10.

**Presentation is sometimes difficult to follow**

– new contributions brought by this study are not clearly stated in abstract, introduction and conclusions. Reading the manuscript, the main results appear to be: raw radar QPE provides unreliable analyses and bias adjustment is needed; differences are observed between at-site analyses from radar and gauges, but radar analyses lie within the gauge confidence intervals; regionalization approaches provide improved analyses. These results were already reported in literature (see for example Overeem et al, 2009; Marra and Morin, 2015; Panziera et al, 2016; Peleg et al 2017; Marra et al., 2017). In my opinion the study brings a lot of new to the field, in particular the use of (i) new methods (i.e. a peak over threshold approach, QQ plot regression) and (ii) new regionalization approaches for rainfall frequency analyses from remote sensing data sets. Abstract, introduction and conclusions need to be reorganized in these terms, even though the results reported by the authors definitely deserve to be mentioned;

We think that the quality of our datasets can be seen as an improvement compared to previous studies. Thank you for pointing out clearly the originality of our methods in the study. We agree that this originality did not appear explicitly. The text has been substantially improved.

– the presentation of the gauge networks in [2, 20-25] and in section 2.1 is difficult to follow, I recommend reorganizing and rephrasing these parts (how many networks are used?, why are they considered separately?, what are the differences? What the advantages of including each of them? Why not using them together?);

The gauge networks used in the study are presented in section 2.1. The rationale for using them is presented in section 2.2. Additional information have been added on page 6, line 32.

– 4, 8-9: please provide information about these methods and move the reference to Goudenhoofdt and Delobbe, 2016 earlier in the text;

Please see from page 5, line 22.

- organization of the radar datasets (QPE, CAP,...) need to be made clearer (sections 2.2, 2.3);

  These sections have been clarified.

- section 3.1: is difficult to follow; in particular [6, 4-9] and [6, 19-21] are not clear to me;

  Those parts have been rewritten (from page 7, line 25 and from page 8, line 21 ).

- section 3.2: what does "problematic events" mean? How are they identified?; why did the authors focus on 10 extremes?

  This has been clarified in the text. Please see from page 8, line 27.

- 8, 21-22: why mentioning the index flood approach? Here the shape is actually assumed uniform (by the use of the exponential distribution), but I guess this is not what the authors mean with 'regionalization';

  We are only reviewing the literature on regionalization. Furthermore the index flood method was proposed with the Gumbel model (Sveinsson et al., 2001).

- how does the 20 km regionalization of the parameters relate to the 10 km and 50 km used in the following parts of the study? how did the authors check/motivate that 20 km "provides a sufficiently large data sample"?

  The decorellation distance (50 km) and the size of the region (20 km) are two different concepts. To avoid confusion the text has been modified from page 11, line 10. The results suggest that the sample is large enough.

- it is not clear whether the method by Reed et al., 1999 is the one the authors used in this paper;

  The reference has been dropped.

- 9, 28-29: did the authors check for non-stationarity in the data (e.g. changes in the instrumentation, or other)?

  As stated in the text, there are no instrumentation changes. The annual maxima have been found stationary (Vannitsem and Naveau, 2007). No statistical test for the stationarity of peaks over threshold timeseries have been done since it is beyond the scope of this study.

- 11, 14-15: is this expected?

  Not necessarily. Important implications have been derived, thank you. Please see from page 12, line 14.

- 11, 24-25: this problem should be solved by the adopted regionalization (20 km);

  No, it shouldn't. Consider a very intense but super fast storm. The 1-hour accumulation extreme will be overestimated. Please see page 14, line 26.

- 1, 11-13; 4, 8; 4, 20-24; 6, 19-21; 7, 10-11; 9, 1-5; 9, 11-16; 9, 31-32: please rephrase;

  See previous responses. See also page 1, line 11; page 9, line 15; page 12, line 3; and page 12, line 21

**Minor comments**

– abstract: the text of the abstract needs to be better organized;

The abstract has been modified and we think the structure is now more clear.

– page 1, line 3: "independent sliding 1h and 24h rainfall": this is not clear;

Made clearer.

– 1, 9: natural rainfall variability in combination with short record lengths is also to be mentioned as a cause of the mismatch between point and areal frequency analyses (see Peleg et al., 2017 and major comment above);

Please see response above.

– 1, 11: "assuming that the extremes are correlated": this is not clear to me, I guess it is related to the regionalization, but needs to be better written;

It has been removed.

– 1, 18: please remove "very" and "very"; please add a comma after "activities"; please provide a reference for this sentence;

A reference has been added. Please see page 2, line 2.

– 1, 19: sewer systems are an example, but I'd insert an example from other applications, such as dams design/management; [1, 21] sewer systems are usually designed for relatively short return periods, applications requiring long return periods are dams, bridges, etc.;

Done.

– 1, 20: no need to specify "a branch of statistics";

Indeed.

– 2, 3: an example of what the authors mean with "high-resolution" would be helpful for the reader;

This is now specified in the paragraph.

– 2, 5-6: "the best potential is provided by radar QPE". Satellite products are fruitfully being used too and are, often, characterized by longer records. This sentence should be motivated and supported by references;

Please see page 3, line 1.

– 2, 11-12: the reference to Saito and Matsuyama, 2015 looks unrelated to the rest of the text, can you provide some information on its relevant parts;

Done.

– Figure 1: what do the areas in the figure represent? Are they catchments? Are they used in the manuscript?

These are the provinces of Belgium. They are not used but are of interest for climatological purposes.

– 5, 5-6: "The hourly bias obtained... convective storms" can be removed;

We think it is relevant.

– 5, 6-10: is this done with a moving window? Or on 24 h blocks?

The term "sliding" means that we are using a moving window.

– 5, 10-11: Marra and Morin, 2015 quantified this uncertainty;

The reference has been added.

– Table 1: what is the meaning of the "Avail. All" column? Does it mean that "Both" were available?

Yes. This has been clarified.

– 6, 22 and 7, 25: I'd suggest to change these titles to something focusing on the tested product rather than on the rain gauges against which it is compared;

Good idea. The titles have been changed.

– 11, 18-19: the authors may want to check Avanzi et al., 2015 for additional inputs;

Thank you for the suggestion.

– 12, 6: since the work by Frederick et al., 1977, a number of papers are available on the derivation of ARFs from radar data (
[revised manuscript text omitted]

---

## Author Comment (AC2) · 20 Jun 2017

**Authors response to Referee 2 comments**

In this paper a peak-over-threshold method is used to perform an extreme rainfall analysis and to derive return levels from weather radar and rain gauges in Belgium. The importance of this work is high, as radar archives are nowadays long enough to permit the development of extreme rainfall analyses which are of fundamental importance for many applications, but the common annual maxima approach needs even longer time series. However, some important explanations and discussions, in addition to those already highlighted by the first review by F. Marra, are missing in the manuscript, and need to be provided before the article can be accepted for publication.

We would like to thank Luca Panziera for underlying the importance of our work. The detailed comments allowed us to improve the presentation and the discussion of our results.

**Major comments**

**What is new?**

It is somehow difficult to understand which new contribution this paper brings with respect to previous studies, and I think that this should be better highlighted in the text. To my understanding, the main novelty of this paper is the use of a POT method for an extreme rainfall analysis for weather radar data.

As pointed out by the first reviewer, the originality of our work did not appear clearly. The use of the POT method together with the other novelties of our approach are now highlighted in the text. Please see page 3, line 30.

**Temporal Declustering**

As rainfall data need to be declustered in order to remove the temporal correlation in the time series before GPD parameters estimation, the authors choose an interval of 12 hours (for 1-hour rainfall) and 3 days (for daily rainfall) in order for two threshold exceedances to be considered as independent. The choice of these intervals, which should be referred to as run length or run parameters according to the literature, seems reasonable, but it could potentially have a big impact on the derived return levels, as it shapes the exceedances time series whose maxima are used for the parameters estimation. If the data are temporally clustered, such temporal lags could not be long enough to remove dependency, but if the temporal clustering occurs rarely, they could actually lead to a significant bias of the return levels estimates. What do the authors mean as temporal independence? How did the authors choose such temporal lags? Did the authors investigate the effect of changing these values on the parameters estimation and final return levels? The subjective choice of these values should be motivated and discussed in the text.

How to deal with declustering is indeed a crucial point to address in extreme rainfall analysis. It is now properly discussed from page 7, line 25.

**Exponential distribution**

As the choice of a null shape parameter is fundamental for this work, I think that it should be motivated more in the text. Therefore, I suggest to briefly report and discuss the main results of Willems (2000), in order to better understand the motivation of this choice The text states also that this choice was taken because of the short period. However, with a POT approach the shortness of the period should not be a limiting factor, as many events are considered. It should also be discussed if this is the best choice for both 1-hour and 24-hours accumulations. Did the authors try to estimate also the shape parameters, to see if from the data a value different from 0 could be derived?

The short period remains a limiting factor to model the tail of the GPD. The choice of the Exponential distribution is further justified from page 8, line 3. We therefore did not try to estimate the shape parameter.

**Radar and gauge comparison**

The authors present an interesting comparison between the radar and gauges extremes, for 1-hour and 24-hours accumulations. Despite this being very interesting and instructive, the implications for this study are not very clear. I suggest the authors discuss at least qualitatively the influence of this investigations on the overall results of the study. The implications of the radar and gauge comparisons have been added from page 9, line 16 and from page 10, line 18.

**Regional frequency analysis**

The regional frequency analysis needs also to be better explained and the choices which were taken need to be motivated and discussed. How did the authors choose the 20-km radius for the analysis? How the resulting return levels at a given pixel should be interpreted, as they stem from exceedances in rainfall values which were observed all around it? Does it still make sense to speak about point measurement? How are the maps of GPD parameters affected by the choice of the 20-km radius circles?

Our methodology should be better explained indeed. An extended literature review is given from page 2, line 24. Our methodology is discussed from page 11, line 5. We think this explains why we can still speak about point measurements. For the derivation of spatial maps, please see from page 12, line 29.

**Return levels maps**

I guess the final goal of the study is to derive maps of return levels with relative uncertainty for Belgium. Despite the return levels are shown for given rain gauge locations, it would be desirable to show also maps of return levels for selected return periods. Would it be possible to insert a map or two of the return levels? How would those maps be affected by the 20-km radius selected for the regional frequency analysis? How these maps should be interpreted? Since you are using a constant shape parameter (equal to 0), and the longest return levels are shaped by it, long return periods map will tend to produce maps less variable in space with respect to short return periods. This should be discussed in the text.

Two return level maps have been added and discussed. Please see page 13, line 4.

**Minor comments**

1. The title is rater general, and you might want to consider adding the name of the region for which this study was performed (Belgium)

   Good suggestion.

2. In the introduction some relevant papers are missing. I strongly encourage the authors to discuss also the papers referenced by F. Marra in his review.

   The papers are now discussed in the text.

3. Pag.2, line22."in this study, we want to demonstrate the potential of this radar-based QPE to derive point rainfall statistics". I don't think that the aim of this study is this, as the radar pixel does not represent point rainfall statistics. As the authors know, the radar rainfall estimate comes from the reflectivity measured within the sample volume, representing an area- not a point- measurement. The intrinsic difference among radar and gauges measurements should be at discussed in the paper, since a comparison between rain gauges and radar return levels is performed (see also major comment 1 by F. Marra).

   The text has been adapted to reflect this important fact. Please see page 3, line 4.

4. Pag.3, line 4: is there a reference for the 5-10% rain gauges underestimation?

   The reference is the one mentioned above in the text.

5. Pag.3, line 7: improve English. I propose to change "very high" with "10-min" temporal resolution (and delete "10-min accumulations are available from the database")

   Done.

6. Pag. 4, line 25: please clarify the last sentence of section 2.2 which, in its present form, it is not correct. Could change " In addition, the increasing radar sample volume will give lower extreme values" to "In addition, the increasing radar sample volume will produce an underestimation of local small-scale extremes".

   Your suggestion has been integrated.

7. Pag. 5, line 24. First two sentences of section 3.1 need to be reformulated as they are very colloquial.

   The sentences have been reformulated in the introduction.

8. Page 6, line 14. With this method of regression in QQ plots, is there a risk of over fitting? Could you please comment on that?

   We don't think there is a risk of overfitting since we are using a simple exponential model.

9. pag.7, line 13-14 and pag.8 lines 9-10. How this percentage would vary by changing the temporal lags considered for independence? (see major comment 2). "This is what we expect from ....". According to which theory/observations? Please clarify and give references.

Please see the response to major comment 2. The sentence has been reformulated on page 9, line 23.

10. Pag. 8, lines 21-28. It would be more appropriate move the literature review to the Introduction, instead of leaving it in this Methodology section.

This part has been moved to the Introduction.

11. Pag. 8, second paragraph of section 4.1: please clarify the explanation of the regional frequency analysis. Given that your circle has a radius of 20 km, what is the aim of considering all the events within a 50 km radius dependent? Isn't this the same as taking just the max value within the 20-km radius? In case it is, wouldn't be easier just say that you take this maximum within the 20-km radius circle?

The decorrelation distance (50 km) and the size of the region (20 km) are not directly related. But the former implies that all extremes observed within the region are independent. We acknowledge that the explanation was a bit confusing. It has been reformulated from page 11, line 10.

12. Pag.10, lines 6-9. Also here I suggest to move the references to other studies in the Introduction.

This part has been moved to the Introduction.

13. Pag. 10, line 13: "a few pixels having too much (50) .... removed" . This sentence is rather unclear, and this seems a rather subjective choice which can hardly be motivated.

This part has been reformulated from page 12, line 31.

14. Figure 2. I suggest to rename "Extreme 1-hour precipitation quantiles" to "1-hour return levels", to be consistent with theory and common nomenclature in the field.

The figures legends have been modified.

15. Tables 2 and 4. I actually miss how the events in the tables are ordered, if there is a logic.

This is now explained in the table description.

16. Figures 1, 6 and 7: a scale in km would help the interpretation of the figure, for those who are not familiar with Belgium

We added a 100 km circle to the maps.

[revised manuscript text omitted]

---

## Author Comment (AC3) · 20 Jun 2017

**Authors response to Referee 3 comments**

**General comments**

The authors apply local and regional frequency analysis (RFA) for extreme rainfall on two radar data products (advanced QPE and basic CAP) for Belgium and compare the results with station based extreme value statistics. They find that the basic radar product shows unrealistic high extremes, the 24h extremes need bias correction and that the fit of the QPE probability distribution is within the confidence interval of the point distribution. The results for RFA are more complex. The topic of the paper is very important and of high relevance for the community. The results are interesting. However, the description of methodology is not clear enough to follow the procedures and understand all the results. This concerns especially the sampling strategy for RFA. Also the presentation of results could be more distinct. Details are given below. However, the research is worth of publication after the authors have the opportunity to make some revisions.

The authors would like to thank Referee 3 for his encouraging comments and suggestions to improve the paper.

**Minor comments**

1. Abstract, lines 10-15: I cannot really understand these sentences: RFA within 20 km?, which region(s)?, rain gauge vs. automatic gauge?, which radar product?, etc.

   The sentences have been reformulated. Please see from page 1, line 10.

2. Page 6, lines 23ff: It is not fully clear if the 10 highest gauge extremes or the 10 highest radar extremes are selected. In the abstract "rain gauges and collocated radar estimates" is mentioned, so I assume the highest gauge values with collocated radar data are used. This should be stated clearly here in the text as well. The rational for this choice should also be discussed.

   This has been reformulated from page 8, line 27.

3. Page 7, lines 26ff: see comment 2

   This has been reformulated from page 10, line 7.

4. Page 8, section 4.1: The sampling for RFA is not clear to me. Do you do a separate RFA for each 20 km radius? How can you apply a minimum distance of 50 km to secure independence with a 20 km radius? If you apply RFA for each radar pixel and consider a minimum distance of e.g. 10 km, then the (collocated) sample is different for each estimate? What about the "index rainfall"? How did you regionalise it? etc.

   The decorrelation distance (50 km) and the size of the region (20 km) are not directly related. But the former implies that all extremes observed within the region are independent. The sample is indeed different for each target location. Since we consider that the extreme statistics are the same for the region, no "index rainfall" is used. The section has been rewritten for the sake of clarity.

5. Page 10, lines 11-12: ".. using a radius of 10 km (with a decorrelation distance of 50 km)" I don't understand this.

   This means that all pixels in the region are considered spatially dependent. To avoid confusion, the reference to the theoretical decorrelation distance (50 km) has been dropped. Please see page 12, line 30.

6. Fig. 2-5: The many lines in in these figures are hardly to disentangle visually. I have not really a good idea what to do here, may be showing only two distributions with confidence limits or excluding the confidence limits of the radar data, or showing additionally bar plots with a comparison of selected quantiles, etc.?

   We acknowledge that these figures contain a lot of lines but we don't see directly how to simplify them while keeping the essential information.

[revised manuscript text omitted]

---

## Referee Comment (RC4) · F. Marra (Referee) · 21 Jun 2017

Dear authors,
thank you for the exhaustive reply. I am well satisfied by your responses and updates to the manuscript, and I think it is now significantly improved with respect to the original one. However, I am still not convinced by one point in the "interpretation of some results" (point 3 of my review letter).
You confirm your interest in *"the extreme at a given location"* rather than *"the regional maximum extreme"*, but did not actually answer the comment. The proposed edit is: *"We also consider that the extremes observed within the 20 km radius during a time*

[Figure]

*window of 12 h are dependent. As in Wright et al. (2014b), we keep only the maximum amongst dependent values. We therefore implicitly assume that the regional maximum follows the same distribution as the local extremes"*, but my concern is related to the use of maximum values within 20-km range areas rather than to the independence of the extremes.

Using the maximum values within 20-km range areas provides the probability of exceeding a given value in any of the 1-km pixels within the area ($\sim$1250 km$^2$); this probability can be significantly different from the probability of exceeding that particular value in a single given 1-km pixel – i.e. the extreme at a given location – even if the mentioned implicit assumption holds. I think this point still needs to be discussed and dealt with.

Kind regards,

Francesco Marra

---

## Author Comment (AC4) · 23 Jun 2017

As pointed out by the reviewer there is fundamental difference between the probability that a given value is exceeded in any of the 1-km pixels within the 20-km radius area (statistics of regional maximum extremes) and the probability that a given value is exceeded at a given location within that area (statistics of extremes at a given location). In this study we are using the regional maximum peaks to derive statistics of extremes at a given location. If the goal was to obtain statistics on regional maximum extremes, we would have taken 10 years as the effective length of the timeseries (i.e. the length of the radar dataset). Our goal is to obtain the probability of exceeding a value for a given

location in the region and, therefore, we use an effective length based on the number of pixels within the area and the number of independent peaks. This length is much larger than 10 years and gives realistic return period estimates. It is directly related to the average over all pixels of the mean number of exceedance per year. That 's why our approach is similar to the one of Wright et al., 2013. More advanced approaches to study spatio-temporal extremes can be considered but these are beyond the scope of the present study.
* * *

---

## Referee Comment (RC5) · F. Marra (Referee) · 24 Jun 2017

Dear authors,
I misinterpreted this step of the methodology and, after this clarification, fully agree on the interpretation of the results. This is an interesting contribution to the methodologies currently used for rainfall frequency analysis from remote sensing data sets.
Best regards,
Francesco Marra

---

## Referee Comment (RC6) · L. Panziera (Referee) · 4 Jul 2017

**Authors response to Referee 2 comments**

In this paper a peak-over-threshold method is used to perform an extreme rainfall analysis and to derive return levels from weather radar and rain gauges in Belgium. The importance of this work is high, as radar archives are nowadays long enough to permit the development of extreme rainfall analyses which are of fundamental importance for many applications, but the common annual maxima 5 approach needs even longer time series. However, some important explanations and discussions, in addition to those already highlighted by the first review by F. Marra, are missing in the manuscript, and need to be provided before the article can be accepted for publication.

We would like to thank Luca Panziera for underlying the importance of our work. The detailed comments allowed us to improve the presentation and the discussion of our results.

Thank you a lot for the work you have done to improve the clarity of this manuscript. The text, thanks also to the comments of the other reviewers, is now much clearer. I still have a few comments regarding this work.

**Replies to original major comments**

**What is new?**
It is somehow difficult to understand which new contribution this paper brings with respect to previous studies, and I think that this should be better highlighted in the text. To my understanding, the main novelty of this paper is the use of a POT method for an extreme rainfall analysis for weather radar data.

As pointed out by the first reviewer, the originality of our work did not appear clearly. The use of the POT method together
with the other novelties of our approach are now highlighted in the text. Please see page 3, line 30.

Thank you.

**Temporal Declustering**
As rainfall data need to be declustered in order to remove the temporal correlation in the time series before GPD parameters estimation, the authors choose an interval of 12 hours (for 1-hour rainfall) and 3 days (for daily rainfall) in order for two threshold exceedances to be considered as independent. The choice of these intervals, which should be referred to as run length or run parameters according to the literature, seems reasonable, but it could potentially have a big impact on the derived return levels, as it shapes the exceedances time series whose maxima are used for the parameters estimation. If the data are temporally clustered, such temporal lags could not be long enough to remove dependency, but if the temporal clustering occurs rarely, they could actually lead to a significant bias of the return levels estimates. What do the authors mean as temporal independence? How did the authors choose such temporal lags? Did the authors investigate the effect of changing these values on the parameters estimation and final return levels? The subjective choice of these values should be motivated and discussed in the text.

How to deal with declustering is indeed a crucial point to address in extreme rainfall analysis. It is now properly discussed from page 7, line 25.

Thank you for the explanations. My only comment is the following: at pag. 7 line 29 you state that using 3 days temporal lag hardly affects the selection of the 1 hour extremes. Could you please provide some numbers (not necessarily you have to add them in the text)? Did you also try to select a temporal lag shorter than 12 hours (e.g. 6 hours) or longer than that (e.g. 24 hours)? You should also investigate the effects of the different temporal lags on the GPD parameters estimation. You might want not to insert this in the paper, but I would appreciate to see how the GPD parameters change.

**Exponential distribution**
As the choice of a null shape parameter is fundamental for this work, I think that it should be motivated more in the text. Therefore, I suggest to briefly report and discuss the main results ofWillems (2000), in order to better understand the motivation of this choice The text states also that this choice was taken because of the short period. However, with a POT approach the shortness of the period should not be a limiting 5 factor, as many events are considered. It should also be discussed if this is the

best choice for both 1-hour and 24-hours accumulations. Did the authors try to estimate also the shape parameters, to see if from the data a value different from 0 could be derived?

The short period remains a limiting factor to model the tail of the GPD. The choice of the Exponential distribution is further justified from page 8, line 3. We therefore did not try to estimate the shape parameter.

Thank you.

**Radar and gauge comparison**

The authors present an interesting comparison between the radar and gauges extremes, for 1-hour and 24-hours accumulations.
Despite this being very interesting and instructive, the implications for this study are not very clear. I suggest the authors discuss at least qualitatively the influence of this investigations on the overall results of the study.

The implications of the radar and gauge comparisons have been added from page 9, line 16 and from page 10, line 18.

Pag. 9, from line 17, you state that due to the randomness of the cell boundary effect, this should not affect the results. And also that the missing data are expected to have only a minor impact on the statistics. I am not totally sure about this, but probably there is not much to do to avoid these problems. You might want to mention this in the text.

**Regional frequency analysis**

The regional frequency analysis needs also to be better explained and the choices which were taken need to be motivated and discussed. How did the authors choose the 20-km radius for the analysis? How the resulting return levels at a given pixel should be interpreted, as they stem from exceedances in rainfall values which were observed all around it? Does it still make sense to speak about point measurement? How are the maps of GPD parameters affected by the choice of the 20-km radius circles?

Our methodology should be better explained indeed. An extended literature review is given from page 2, line 24. Our methodology is discussed from page 11, line 5. We think this explains why we can still speak about point measurements. For the derivation of spatial maps, please see from page 12, line 29.

Thank you for the better explanation.

**Return levels maps**

I guess the final goal of the study is to derive maps of return levels with relative uncertainty for Belgium. Despite the return levels are shown for given rain gauge locations, it would be desirable to show also maps of return levels for selected return periods. Would it be possible to insert a map or two of the return levels? How would those maps be affected by the 20-km radius selected for the regional frequency analysis? How these maps should be interpreted? Since you are using a constant shape parameter (equal to 0), and the longest return levels are shaped by it, long return periods map will tend to produce maps less variable in space with respect to short return periods. This should be discussed in the text.

Two return level maps have been added and discussed. Please see page 13, line 4.

Thank you for adding the maps of the return levels. if I am correctly interpreting the maps, I think that they are strongly influenced by the selected circle for the RFA, and  I think that this does not appear clearly from the discussion of page 13, lines 4-21. I see this as a major drawback of the proposed method, and, this should be clearly mentioned in the text.

Do you also have maps for the 24 hours RFA? Can you show them as well?

**Other comments**

1. Pag. 2 line 16: delete the parenthesis from the citation Willems et al. (2007)

2. Pag. 2 line 32: I would delete "perfectly": even for widespread rainfall situation, local phenomena can lead to gradients in rainfall rarely measured by the rain gauges.
3. Pag. 3 line 5 at the end: "It has been shown", please correct.
4. Pag. 3, line 31: to be correct, also the work of Panziera et al. (2016) used a radar QPE combined with rain gauges. Are you sure that all the other mentioned papers did not also consider a combination of radar and rain gauges?
5. Pag. 8, line 8. Why the EXP model is less affected by observational errors?
6. Table 2: the event 7 should not be the 6[th], as 25.17 (radar, event 7) is larger than 24.35 (radar, event 6)?
7. Pag. 11, line 14. I don't think that mentioning here my 2016 paper is required.
8. Pag. 13, line 13.I think that the hail contamination is more sensitive to the radar sampling volume than the height of the beam, since a larger sample volume means also a smaller probability that it is dominated by hail, but also other hydrometeors types are present. I think that hail contamination is more relevant close to the radar.
9. Pag. 14, line 23-32 and 34-35 look like a plan for future work, so you might consider to move them at the end of the Conclusions. I also suggest to reorganize the Conclusions sections, as methodological information and the results are mixed together making this sections not easy to read.

---

## Author Comment (AC5) · 4 Jul 2017

Thank you for considering our response and your additional comments on the revised manuscript.

Temporal Declustering

Choosing 3 days instead of 12 hours for the temporal lag removes rank 30 (radar) and rank 14 (gauge) at station Humain ; it removes rank 27 (gauge) at station Uccle. This changes very slightly the scale parameter but only for the gauge : from 7.5 to 7.6 and

from 6.8 to 6.9, respectively. Using 6 hours instead of 12 hours does not change the extremes up to rank 30 for the radar and the gauge at the two stations.

Radar and gauge comparison

The text has been clarified as followed : "Since the level of missingness is limited, the impact on the statistics is expected to be small".

Return levels maps

There is indeed an impact of the circle from the RFA but it is relatively limited in most of the study area. Since this is mainly due to radar artifacts we don't consider it as a drawback of the proposed RFA method. The discussion has been improved as follows : "Circular patterns appear on the maps due to the influence of the pixels located at their centers. The high values are caused by pixels contaminated by non-meteorological echoes (e.g. at the German border) and hail. A stronger filter for non-meteorological echoes is not used because it could remove actual precipitation information. The circular effect might be reduced by using a larger radius or a higher threshold rank but this is computationally expensive." The RFA has been limited to 1 hour extremes in this paper since it has the best potential for radar data. Extending the approach to other durations is interesting for future research.

Other comments

1. Done.

2. Done.

3. Done.

4. We do not refer to radar and gauge merging. We mean a quality similar to our datasets : reanalysed and verified radar-based QPE (with or without gauge merging) and as reference 10 min quality-controlled rain gauge data with 40 years of records.

5. Since the GPD has one more parameter than the EXP, it will react more to individual errors in the data.

6. This has been corrected.

7. The reference has been dropped.

8. Due to the significantly higher reflectivity of hail, the averaged value from a large sample volume should still exceed the hail threshold of 55 dBZ. The probability of very high reflectivity is believed to increase with altitude due to the dynamics of convective storms and hail processes.

9. The Conclusions have been organised in two sections :  "Results" and "Prospects".  We think it is relevant to combine the methodological information and the results in the Conclusions.

---

## Referee Comment (RC7) · L. Panziera (Referee) · 5 Jul 2017

Dear authors, thank-you a lot for the replies to my comments. I think that the paper is now more complete and suitable for publication. Luca Panziera
* * *